# Molecular function limits divergent protein evolution on planetary timescales

Mariam M Konaté[1,2†], Germán Plata[1†*], Jimin Park[1,3], Dinara R Usmanova[1], Harris Wang[1,3], Dennis Vitkup[1,4*]

[1]Department of Systems Biology, Columbia University, New York, United States; [2]Division of Cancer Treatment and Diagnosis, National Cancer Institute, Bethesda, United States; [3]Department of Pathology and Cell Biology, Columbia University, New York, United States; [4]Department of Biomedical Informatics, Columbia University, New York, United States

**Abstract** Functional conservation is known to constrain protein evolution. Nevertheless, the long-term divergence patterns of proteins maintaining the same molecular function and the possible limits of this divergence have not been explored in detail. We investigate these fundamental questions by characterizing the divergence between ancient protein orthologs with conserved molecular function. Our results demonstrate that the decline of sequence and structural similarities between such orthologs significantly slows down after ~1–2 billion years of independent evolution. As a result, the sequence and structural similarities between ancient orthologs have not substantially decreased for the past billion years. The effective divergence limit (>25% sequence identity) is not primarily due to protein sites universally conserved in all linages. Instead, less than four amino acid types are accepted, on average, per site across orthologous protein sequences. Our analysis also reveals different divergence patterns for protein sites with experimentally determined small and large fitness effects of mutations.

**Editorial note:** This article has been through an editorial process in which the authors decide how to respond to the issues raised during peer review. The Reviewing Editor's assessment is that all the issues have been addressed (see decision letter).

DOI: https://doi.org/10.7554/eLife.39705.001

*For correspondence:
gap2118@cumc.columbia.edu
(GáP);
dv2121@cumc.columbia.edu (DV)

†These authors contributed equally to this work

Competing interests: The authors declare that no competing interests exist.

## Introduction

As proteins evolve from a common ancestor, their sequences and structures diverge from each other (*Chothia and Lesk, 1986*; *Povolotskaya and Kondrashov, 2010*). Multiple previous studies have investigated the relationship between the conservation of protein molecular function, sequence identity (*Lee et al., 2007*; *Tian and Skolnick, 2003*; *Worth et al., 2009*) and structural similarity (*Chothia and Lesk, 1986*; *Wilson et al., 2000*). For example, the likelihood that two proteins share the same molecular function, given their sequence (*Tian and Skolnick, 2003*) or structural (*Wilson et al., 2000*) similarity, has been used to investigate the emergence of new protein functions (*Rost, 2002*; *Conant and Wolfe, 2008*), and to perform functional annotations of protein sequences (*Lee et al., 2007*; *Wilson et al., 2000*). In this work, we focused on a different and currently unaddressed set of questions. Namely, how far can two sequences diverge while continuously maintaining the same molecular function? What are the temporal patterns of this divergence across billions of years of evolution? And how different protein sites contribute to the long-term divergence between orthologs with the same molecular function? We note that the requirement for the continuous conservation of molecular function is crucial in this context, as multiple examples of convergent evolution and protein engineering demonstrate that the same molecular function, such as catalysis of a

specific chemical reaction, can in principle be accomplished by proteins with unrelated sequences and even different folds (*Bork et al., 1993*; *Galperin et al., 1998*; *Omelchenko et al., 2010*).

It was previously demonstrated that proteins with the same structural fold frequently diverge to very low (~10%) levels of sequence identity (*Rost, 1997*). These results suggest that conservation of protein fold, that is, the overall arrangement and topological connections of protein secondary structures (*Murzin et al., 1995*), exerts relatively minor constraints on how far protein sequences can diverge. In contrast to protein folds, it is possible that conservation of specific molecular functions will significantly limit the long-term divergence of protein orthologs. While only a relatively small fraction of protein residues (~3–5%) are usually directly involved in catalysis (*Lehninger et al., 2013*), recent analyses have demonstrated that even sites located far from catalytic residues may be significantly constrained in evolution. Because substitutions at these sites can have substantial effects on molecular function (*Firnberg et al., 2016*), it is likely that functionally-related sequence constraints extend far beyond catalytic residues.

In this study, we explored the long-term divergence patterns of protein orthologs by characterizing their pairwise sequence and structural similarity as a function of their divergence time. We used several models of molecular evolution to calculate the divergence rates, defined as the decrease in pairwise sequence identity or structural similarity per unit time, between orthologous proteins with the same molecular function. We also characterized the long-term divergence patterns at protein sites with different levels of evolutionary conservation, different locations in protein structures, and different experimentally measured fitness effects of amino acid substitutions. Finally, we explored how the limits of sequence and structural divergence after billions of years of evolution depend on the degree of functional conservation between orthologs.

## Results

To study the evolution of proteins with the same molecular function, we initially focused our analysis on enzymes because their molecular function is usually well defined. The Enzyme Commission (EC) classifies enzymatic functions using a hierarchical 4-digit code (*Bairoch, 1999*), such that two enzymes that share all four EC digits catalyze the same biochemical reaction. We used protein sequences representing 64 EC numbers from 22 diverse model organisms across the three domains of life (*Supplementary file 1*). The considered activities include members of all six major enzyme classes: oxidoreductases, transferases, hydrolases, lyases, isomerases and ligases.

To investigate whether the conservation of enzymatic function limits the divergence between orthologous sequences, we first calculated global pairwise sequence identities between orthologs as a function of their divergence times (*Figure 1*, *Figure 1—figure supplement 1*). The pairwise divergence times reported in the literature (*Hedges et al., 2006*) between the considered 22 species (*Supplementary file 1*) were used as a proxy for the divergence times between corresponding orthologous proteins. For each enzymatic activity, we constructed phylogenetic trees based on the orthologous protein sequences and compared them to the corresponding species' trees. Protein sequences with clear differences to the species'phylogenetic tree topologies, suggesting cases of horizontal gene transfer, were excluded from the analysis (see Materials and methods).

We next considered two simple models of long-term protein evolution, one without a long-term limit of sequence divergence and the other with an explicit divergence limit. The first model corresponds to sequence divergence with equal and independent amino acid substitution rates across all proteins sites (*Dickerson, 1971*; *Zuckerkandl and Pauling, 1965*); see *Equation 1*, where $y$ represents global sequence identity, $t$ represents divergence time, and $R_0$ represents the average substitution rate (*Dickerson, 1971*). In this model, back substitutions are not allowed, and sequence divergence slows down with time simply due to multiple substitutions at the same protein sites and progressively fewer non-mutated sites. The second model corresponds to sequence divergence where, in addition to sites with equal and independent substitution rates, there is a minimal fraction of identical sites at long divergence times; the fraction of identical sites is represented by $Y_0$ in *Equation 2*.

$$y = 100 * e^{-R_0 * t} \tag{1}$$

$$y = Y_0 + (100 - Y_0) * e^{-R_0 * t} \tag{2}$$

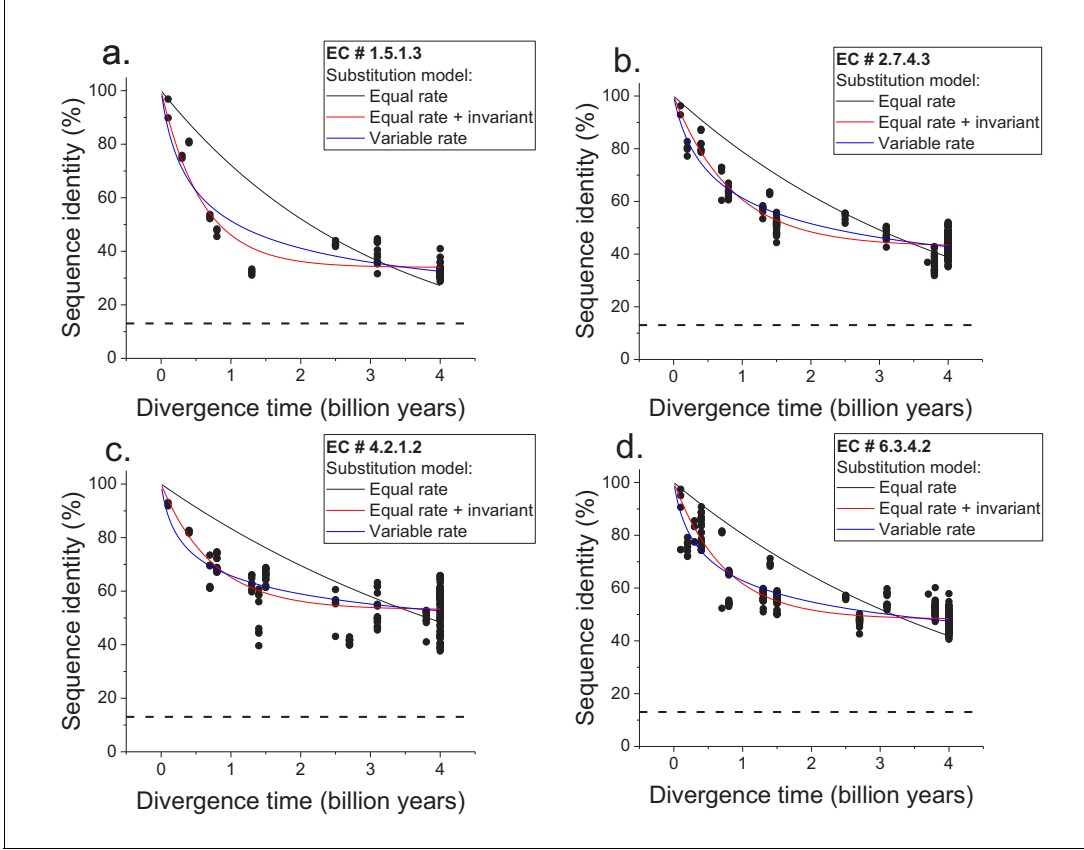

**Figure 1.** Sequence divergence of enzyme orthologs as a function of time. The global pairwise sequence identities between pairs of orthologous enzymes are shown as a function of divergence times between the corresponding species. Three models of amino acid substitution were used to fit the divergence data. Model 1 (black lines) assumes independent and equal substitution rates across all protein sites. Model 2 (red lines) assumes, in addition to independent and equal substitution rates, that a given fraction of protein sites remains identical at large divergence distances. Model 3 (blue lines) assumes a gamma distribution of amino acid substitution rates across sites. Best fits of the models are shown for four representative EC numbers: (a) 1.5.1.3, (b) 2.7.4.3, (c) 4.2.1.2, (d) 6.3.4.2. The horizontal dashed black lines represent the average sequence identity for the global alignment of unrelated protein sequences. The data and corresponding model fits for the other EC numbers considered in the analysis are given in *Figure 1—figure supplement 1* and *Supplementary file 2a*.

DOI: https://doi.org/10.7554/eLife.39705.002

The following source data and figure supplements are available for figure 1:

**Figure supplement 1.** Sequence divergence of enzyme orthologs as a function of time.
DOI: https://doi.org/10.7554/eLife.39705.003

**Figure supplement 1—source data 1.** Sequence identity versus divergence times for 64 enzyme families.
DOI: https://doi.org/10.7554/eLife.39705.004

**Figure supplement 2.** Projected long-term sequence identity of metabolic orthologs.
DOI: https://doi.org/10.7554/eLife.39705.005

**Figure supplement 3.** Divergence of orthologs with experimentally validated functional annotations.
DOI: https://doi.org/10.7554/eLife.39705.006

**Figure supplement 3—source data 1.** Sequence identity versus divergence times for experimentally validated enzymes.
DOI: https://doi.org/10.7554/eLife.39705.007

**Figure supplement 4.** Enzyme divergence rates at 30% sequence identity.
DOI: https://doi.org/10.7554/eLife.39705.008

**Figure supplement 5.** Sequence divergence of non-enzyme orthologs as a function of divergence time.
DOI: https://doi.org/10.7554/eLife.39705.009

**Figure supplement 5— source data 1.** Sequence identity versus divergence times for 29 non-enzyme families.
DOI: https://doi.org/10.7554/eLife.39705.010

**Figure supplement 6.** Sequence divergence of mitochondrial ribosomal orthologs as a function of divergence time.
DOI: https://doi.org/10.7554/eLife.39705.011

*Figure 1 continued on next page*

*Figure 1 continued*

**Figure supplement 6—source data 1.** Sequence identity versus divergence times for mitochondrial ribosomal orthologs.
DOI: https://doi.org/10.7554/eLife.39705.012
**Figure supplement 7.** Effect of uncertainty in the estimation of species divergence times on the model fits.
DOI: https://doi.org/10.7554/eLife.39705.013

We applied the two models to fit the sequence divergence for each of the considered enzymatic functions. The best model fits for four representative metabolic activities are shown in *Figure 1* (black for the first model and red for the second); the fits for the remaining metabolic activities are shown in *Figure 1—figure supplement 1*. In 62 of the 64 cases, the second model fits the divergence data significantly better than the first model (F-test p-value<0.05, *Supplementary file 2a*). Moreover, in 95% of the cases (61/64) the maximum likelihood value of the parameter $Y_0$ is significantly higher (Wald test p-value<0.05) than the average sequence identity between random protein sequences based on their optimal global alignment (~13.5%, shown in *Figure 1* and *Figure 1—figure supplement 1* by dashed black lines). The distribution of the fitted parameter $Y_0$ suggests a long-term sequence identity >25% (with average ~40%) between considered orthologs (*Figure 2a*); this demonstrates that conservation of a specific enzymatic function significantly limits long-term protein sequence divergence. Notably, model two is mathematically equivalent (see Materials and methods) to a divergence model with equal substitution rates across sites, a limited number of amino acid types accepted per site, and allowed back substitutions (*Tajima and Nei, 1984*; *Gilson et al., 2017*; *Yang, 2006*). In this model, the parameter $Y_0$ represents the inverse of the effective number of acceptable amino acid types per site during protein evolution. Our results thus suggest that, on average, only 2 to 4 amino acids are acceptable per site for proteins that strictly conserve their molecular function (*Figure 2a*, top blue X axis); we note that this low average number does not contradict the fact that more than four amino acid types could be observed at a given protein site at low frequencies (*Breen et al., 2012*).

The two aforementioned models simplify the process of sequence divergence by considering the same substitution rates across protein sites. A more realistic and commonly used model of protein evolution assumes a gamma distribution (*Yang et al., 2000*) of substitution rates across protein sites; see *Equation 3*; (*Ota and Nei, 1994*), where α represents the shape parameter of the gamma distribution. The best fits of such a variable-rate model (blue in *Figure 1* and *Figure 1—figure supplement 1*) showed that the rates of protein sequence divergence between orthologous

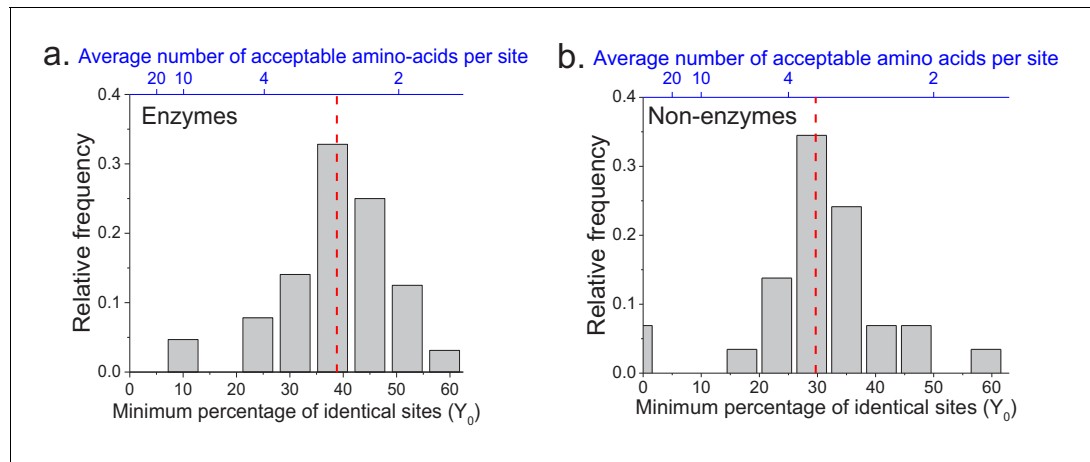

**Figure 2.** The limit of long-term protein sequence divergence between orthologous proteins. (**a**) The distribution of $Y_0$ parameter values across 64 EC numbers for fits based on Model 2 (*Equation 2*). The $Y_0$ parameter represents the minimum percentage of protein sites that remain identical at long divergence times. The parameter $Y_0$ (considered as a fraction) can also be interpreted as the inverse of the average number of amino acid types accepted per protein site during long-term protein evolution (top blue X axis). (**b**) Similar to panel a, but for 29 ancient protein families annotated with non-enzymatic functions. In panels a and b, the vertical red dashed lines represent the median values of the distributions (39% and 30%, respectively).
DOI: https://doi.org/10.7554/eLife.39705.014

enzymes have decreased by more than 10 times during ~4 billion years of evolution (see Materials and methods and *Supplementary file 2b*). Although the third model does not explicitly consider a long-term divergence limit, the obtained model fits also show that the vast majority of orthologous enzymes with the same function will remain above 25% sequence identity on the time-scales when Earth environments will be hospitable to life (1–3 billion years from the present *O'Malley-James et al., 2014*) (*Figure 1—figure supplement 2*).

$$y = 100 * \left( \frac{R_0 * t}{\alpha} + 1 \right)^{-\alpha} \tag{3}$$

The observed divergence limit is not an artifact due to difficulty detecting remote protein homologs, as it occurs at relatively high sequence identities (*Figure 1* and *Figure 1—figure supplement 1*), for which corresponding orthologs can be easily identified by computational sequence comparison methods. Furthermore, the results remained similar when we restricted the analysis to orthologous enzyme pairs with experimentally validated molecular functions (*Figure 1—figure supplement 3*), based on publications referenced in the BRENDA database (*Chang et al., 2015*). The results also remain robust towards the variance in the estimates of divergence times between considered species (see Materials and methods). We note that the divergence limit between orthologs with the same molecular function does not imply that the rates of molecular substitutions decrease in evolution. It is also not simply due to the curvilinear relationship between time and sequence identity caused by multiple mutations at the same sites; specifically, the observed decrease in divergence rates is substantially higher (by >10 fold) than the one expected in model one simply due to multiple substitutions at the same protein sites. Instead, the effective limit is reached when, due to a small number of amino acids types accepted per protein site and back substitutions, additional amino acid replacements do not lead to a substantial further increase in protein sequence and structural divergence (*Meyer et al., 1986*).

Interestingly, following the previously introduced metaphor of the expanding protein universe (*Povolotskaya and Kondrashov, 2010*; *Dokholyan et al., 2002*), we can use the third model (*Equation 3*) to express the divergence rate between orthologs as a function of protein distance ($D = 1 - y$, where $y$ is the fractional sequence identity ranging from 0 to 1), see *Equation 4*. This equation, similarly to Hubble's law of universe expansion (*Hubble, 1929*), describes how the divergence rate depends on the distance between protein orthologs. In contrast to the real universe, the expansion rate of the protein universe significantly decreases with divergence time and with distance between protein orthologs. For example, the divergence rate between protein orthologs drops, on average, to only ~2% sequene identity decrease per billion years when their mutual sequence identity reaches 30% (corresponding to protein distance of 70%; *Figure 1—figure supplement 4*).

$$\frac{\partial D}{\partial t} = R_0 * (1 - D)^{\frac{(\alpha + 1)}{\alpha}} \tag{4}$$

The analyses described above focused on the divergence of enzymes with the same molecular function. In order to investigate whether the observed divergence patterns are not specific to enzymes, we repeated the analysis using non-enzymatic ancient orthologs (*Figure 1—figure supplement 5*, *Supplementary file 2c*). The set of analyzed 29 protein families included ribosomal proteins, heat shock proteins, membrane transporters, and electron transfer flavoproteins (*Supplementary file 2d*). Based on the same set of 22 species used in the analysis of enzyme families, we found that model two fitted the data significantly better than model one, and that the parameter $Y_0$ was >25% for the majority (23/29) of the non-enzymatic protein families (*Figure 2b*, *Supplementary file 2c*). Interestingly, we also identified 19 additional orthologous groups showing two clearly different divergence patterns (*Figure 1—figure supplement 6*), with pairs of eukaryotic orthologs diverging substantially faster and farther than prokaryotic orthologs in the same protein family. The orthologous groups with this behavior included mitochondrial ribosomal proteins and initiation factors of mitochondrial translation (*Supplementary file 2e*). It has been previously postulated that mitochondrial ribosomal proteins diverged significantly faster in eukaryotes, compared to the divergence between their bacterial orthologs, due to compensatory protein substitutions

following the accumulation of slightly deleterious substitutions in the mitochondrial ribosomal RNA (*Barreto and Burton, 2013*).

Having established, in the first half of the manuscript that conservation of molecular function significantly limits long-term sequence evolution, we investigated, in the second half, how different protein sites contribute to the observed divergence constraints. Specifically, whether the same protein sites are conserved between ancient orthologs in different phylogenetic lineages, how sites with different fitness effects of amino acid substitutions contribute to the divergence limit, and how structural locations of protein sites affect their long-term divergence patterns. We also explored how different levels of functional specificity constrain sequence and structural divergence.

To investigate whether the same protein sites are usually conserved between orthologs in different phylogenetic lineages, we aligned the sequences of ancient enzyme orthologs with the same molecular function (see Materials and methods). We then calculated how often each protein site is occupied by identical amino acids across pairs of orthologs from phylogenetically independent linages (*Figure 3—figure supplement 1*). Orthologous protein pairs from independent lineages were obtained using species' pairs that do not share any edges in the corresponding phylogenetic tree (*Arnold and Stadler, 2010*) (*Figure 3a*); for example, in *Figure 3a* the pair D-H is independent of the pair A-B but not of the pair E-F. We performed the above analysis using 30 enzymatic activities for which at least 20 independent pairs of orthologs could be identified based on annotations in the KEGG database (*Kanehisa et al., 2016*) (see Materials and methods). The results demonstrated that only a relatively small fraction of protein sites (10–20%) are universally conserved, that is, they are identical in a majority (>90%) of independent lineages (*Figure 3b*). Therefore, the observed long-term divergence limit is not primarily due to sets of universally conserved protein sites; instead, different sites contribute to the limit in independent phylogenetic lineages. By comparing the fractions of universally conserved sites to the average sequence identity between distant orthologs (~40%, *Figure 2a*) we found that these sites account, on average, for only ~35% of the observed sequence identity at long divergence distances. The analysis also revealed that different protein

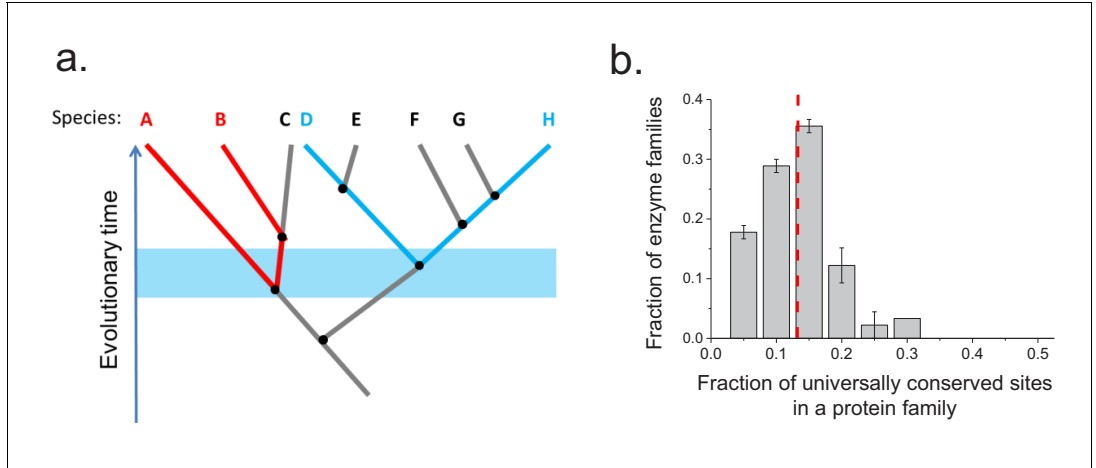

**Figure 3.** Conservation of protein sites in phylogenetically independent lineages. To identify the fractions of protein sites that are universally conserved — defined as sites that are identical in at least 90% of orthologs — we considered phylogenetically independent lineages. (a) Illustration of pairs of species (e.g. A–B and D–H) representing phylogenetically independent lineages. In the figure, A-B and D-H are pairs of species that diverged within a certain time window (illustrated by the blue shaded region); these species pairs do not share more recent edges in the phylogenetic tree. (b) The distribution of the fraction of universally conserved sites across 30 enzymatic families. The analysis was performed using 30 enzymatic families for which at least 20 independent pairs of orthologs with the same molecular function could be identified based on annotations in the KEGG database (*Kanehisa et al., 2016*) (see Materials and methods); pairs of orthologs that diverged >2 billion years ago were selected for this analysis. Error bars represent the S.E.M. based on three replicates using different sets of orthologous pairs. The dashed red line represents the median of the distribution (~13%).

DOI: https://doi.org/10.7554/eLife.39705.015

The following figure supplement is available for figure 3:

**Figure supplement 1.** Distribution of enzyme sites according to their conservation frequency.
DOI: https://doi.org/10.7554/eLife.39705.016

families show different probability distributions of identical sites (*Figure 3—figure supplement 1*). This is likely a consequence of diverse structural and functional requirements across protein families, leading to protein-family specific constraints on protein sites.

We next investigated the long-term divergence patterns at protein sites with different fitness effects of amino acid substitutions. To that end, we experimentally measured the fitness effects of all possible single amino acid substitutions in a representative enzyme, the *Escherichia coli* dihydrofolate reductase (FolA, EC 1.5.1.3). We selected FolA for the experiments due to its small size (159 amino acids) and essential role in the *E. coli* metabolism (*Benkovic et al., 1988*); also, the long-term protein sequence identity between FolA orthologs (~32%, see *Figure 1a*) is similar to other analyzed enzymes (*Figure 2a*). Following a recently described strategy (*Kelsic et al., 2016*), we used the Multiplex Automated Genome Engineering (MAGE) approach (*Wang et al., 2009*) to introduce every possible amino acid substitution at each FolA site in *E. coli*. To evaluate the relative fitness effects of protein substitutions we measured the growth rate of strains containing each protein variant compared to the 'wild type' (WT) strain into which substitutions were introduced. Relative growth rates were measured in parallel by performing growth competition experiments between the pooled mutants. Amplicon sequencing of the *folA* gene was then used to measure the relative changes of mutant and WT abundances as a function of time (see Materials and methods, *Supplementary file 3*).

Using the MAGE growth measurements in *E. coli*, we investigated the patterns of long-term sequence divergence at protein sites with different fitness effects of amino acid substitutions. Specifically, we sorted FolA protein sites into several groups according to their experimentally measured average fitness effects (*Figure 4—figure supplement 1*), and explored sequence divergence for sites within each fitness group (*Figure 4a*, different colors). We evaluated sequence identity between FolA orthologs across divergence times using all pairwise comparisons between ~300 orthologous sequences from the COG database (*Galperin et al., 2015*). Although, as expected, sites with stronger fitness effects diverged more slowly, our analysis revealed interesting differences in temporal divergence patterns for sites with small and large fitness effects. For sites in the least deleterious fitness group (*Figure 4a*, blue) we observed, similar to the global sequence identity, a substantial decrease (~10 fold, see *Equation 5* in Materials and methods) in mutual divergence rates after ~1.5 billion years of evolution. Notably, even for FolA sites with mild fitness effects, sequence identity remains above 25% at long divergence distances. In contrast to sites with mild fitness effects, sites with the most deleterious mutations (*Figure 4a*, black) displayed a much slower, but approximately constant average divergence rate throughout evolutionary history. This pattern suggests that, in contrast to divergence at sites with small fitness effects, the divergence at sites with large effects is not yet close to saturation.

To assess the generality of the FolA results we used another dataset (*Kelsic et al., 2016*), obtained using MAGE, of fitness values for all possible amino acids substitutions in the *E. coli* translation initiation factor InfA (*Figure 4b*). Consistent with the relatively higher level of sequence conservation of InfA, we observed lower average mutant growth rates and lower rates of sequence divergence in each fitness group. Nevertheless, the long-term divergence patterns were qualitatively similar between the two proteins. For sites in the least deleterious InfA fitness group (*Figure 4b*, blue), we observed a substantial decrease in the divergence rate after ~2 billon years of evolution. In contrast, sites with strongest fitness effects (*Figure 4b*, pink) displayed a slower but approximately constant divergence rate.

Because the fitness effects of mutations at a protein site may change in evolution (*Lunzer et al., 2010*; *Chan et al., 2017*), it is interesting to investigate how fitness effects measured in one species, such as *E. coli*, correlate with the site conservation in other species at the divergence limit. To explore this question, we calculated the probability that a protein site is occupied at long evolutionary times (~2 billion years for FolA and ~2.5 billion years for InfA) by the same amino acid in phylogenetically independent lineages (*Figure 3a*). We then investigated how this probability changes as a function of the average fitness effects of substitutions at the site measured in *E. coli* (*Figure 4c*). For both FolA and InfA, the probability that a protein site is identical at large divergence distances first increases, approximately linearly with increasing average fitness effects, and then begins to saturate for sites with large (>30% growth rate decrease) fitness effects. Thus, the fitness effects at a protein site correlate with the site's conservation even after billions of years of evolution.

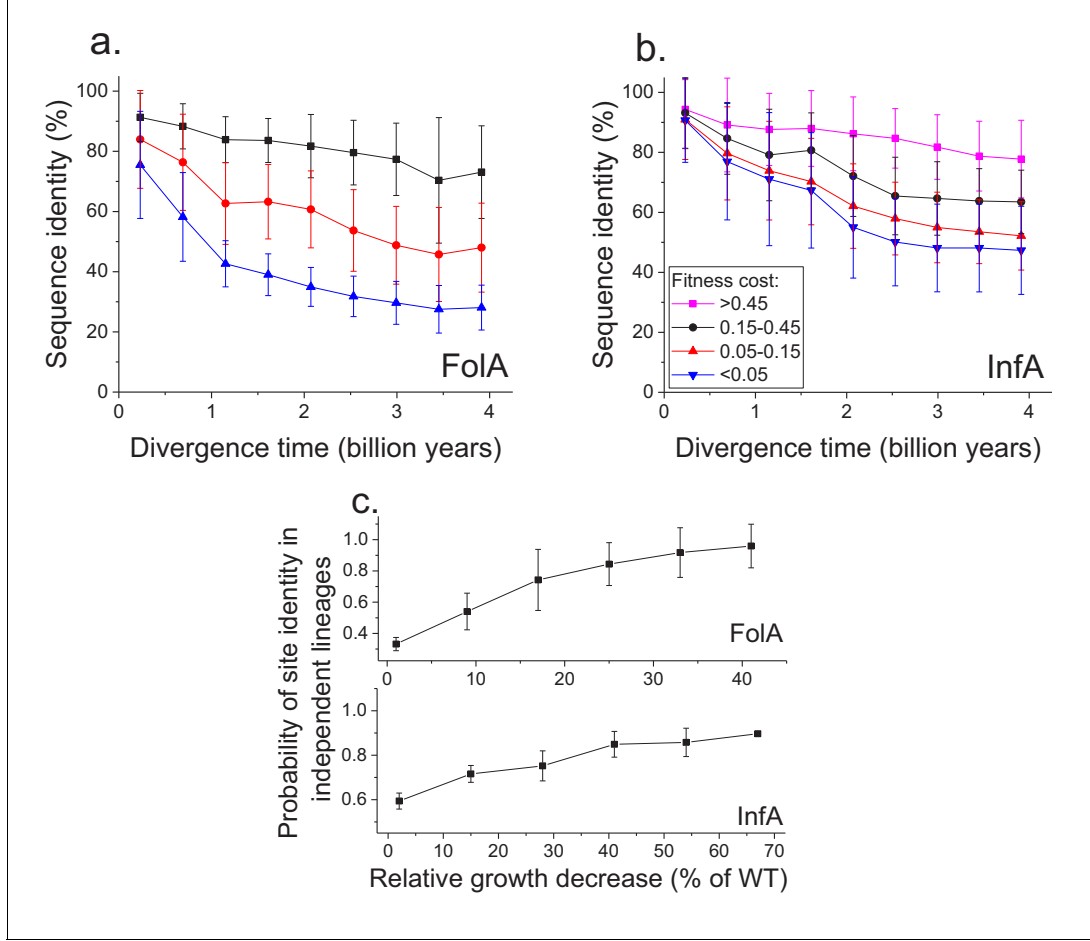

**Figure 4.** Sequence divergence of protein sites with different fitness effects of mutations measured in *E. coli*. (a) The divergence of sequence identity for FolA protein sites with different average fitness effects of mutations (measured in *E. coli*) is shown using different colors. The average sequence identities were calculated using bacterial FolA orthologs available in the COG database (*Galperin et al., 2015*); divergence times were estimated using bacterial 16S rRNA sequences (see Materials and methods). Error bars represent the S.D. of sequence identity in each bin. (b) Similar to panel (a), but for the sequence divergence at protein sites of the *E. coli* translation initiation factor InfA. (c) The probability that protein sites in FolA orthologs (upper panel) and InfA orthologs (lower panel) are occupied by identical amino acids as a function of the average mutant fitness (measured in *E. coli*) at the corresponding protein sites. The probability represents the fraction of phylogenetically independent pairs of orthologs in which sites are identical at long divergence times (2 ± 0.25 billion years for FolA, and 2.5 ± 0.25 billion years for InfA). Error bars represent the S.E.M. across sites.
DOI: https://doi.org/10.7554/eLife.39705.017

The following figure supplements are available for figure 4:

**Figure supplement 1.** Distribution of average fitness effects of amino acid substitutions.
DOI: https://doi.org/10.7554/eLife.39705.018

**Figure supplement 2.** Reproducibility of experimentally measured average fitness effects of amino acid substitutions across FolA sites.
DOI: https://doi.org/10.7554/eLife.39705.019

The sequence constraints revealed by our analysis likely arise due to the conservation of corresponding protein structures required for efficient catalysis and molecular function (*Wilson et al., 2000*; *Watson et al., 2005*). Therefore, in addition to sequence divergence we also investigated the long-term structural divergence of orthologous proteins with the same function. For this analysis we used >1000 orthologous pairs of enzymes sharing all 4 EC digits with known 3D structures (*Berman et al., 2000*) (see Materials and methods); structures of orthologous enzymes were aligned using the TM-align algorithm (*Zhang and Skolnick, 2005*). This analysis demonstrated that the average root mean square deviation (RMSD) between

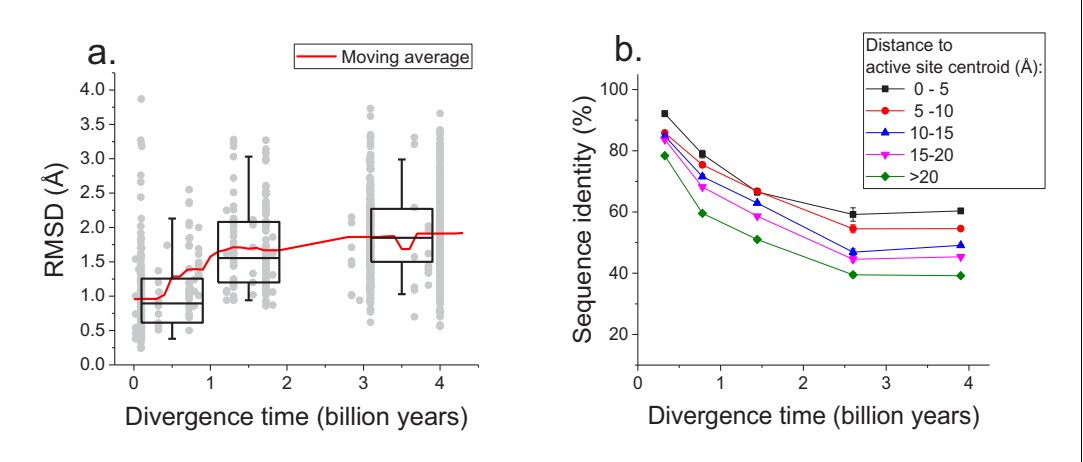

**Figure 5.** Long-term structural evolution of enzymes with the same molecular function. (**a**) The pairwise C-alpha root mean square deviation (RMSD) as a function of the divergence time between pairs of orthologs annotated with the same EC number. RMSD values were calculated based on structural protein alignments using the TM-align algorithm (*Zhang and Skolnick, 2005*). Gray dots represent pairs of considered orthologs, boxes indicate the median and 25–75 RMSD percentiles for the corresponding divergence times, the vertical lines indicate the 5–95 percentiles, and the red line shows the moving average of the data. (**b**) Long-term divergence of sequence identity of protein sites located at different distances to enzymes' active site residues. In this analysis we considered the same species and enzymatic activities used to explore the global sequence divergence (*Figure 1* and *Figure 1—figure supplement 1*); the average sequence identities within each distance shell (shown using different colors) were calculated across all pairs of orthologs annotated with the same EC number (see Materials and methods). Error bars represent the S.E.M. across ortholog pairs.

DOI: https://doi.org/10.7554/eLife.39705.020

The following source data is available for figure 5:

**Source data 1.** RMSD versus divergence times for proteins with the same enzymatic function.
DOI: https://doi.org/10.7554/eLife.39705.021

C-alpha atoms of the orthologous enzymes increases (Spearman's r = 0.44, p-value<1e-20) with divergence time (*Figure 5a*). Nevertheless, the C-alpha RMSD between orthologs rarely increases beyond 3 Å, even at long evolutionary distances. Consistent with sequence evolution (*Figure 1*), we also observed a substantial decrease in the rate of structural divergence after ~1.5 billion years of divergent evolution.

Only a small fraction of all enzyme residues forms an active site and directly participates in catalysis. Therefore, we investigated next how the sequence divergence depends on the spatial proximity of protein positions to active site residues. It was previously demonstrated that evolutionary rates of amino acid substitutions correlate with protein sites' spatial distance to catalytic residues (*Jack et al., 2016*). Notably, differences in short-term evolutionary rates do not imply the existence of a divergence limit, nor do they inform how site-specific divergence patterns correlate with sites' structural locations. Thus, the main goal of our analysis was to investigate the temporal patterns of the long-term divergence, and the effective divergence limit for sites at various distances to the active site. We considered catalytic site annotations available from the Protein Data Bank (*Berman et al., 2000*), UniProt-KB (*UniProt Consortium, 2015*) and the Catalytic Site Atlas (*Porter, 2004*) and quantified the average sequence divergence for sites at various distances from catalytic residues (see Materials and methods, *Figure 5b*). We based this analysis on the same set of enzymatic activities used to study global sequence divergence (*Figure 1* and *Figure 1—figure supplement 1*). Although, as expected, residues close to the active site were the most highly conserved (*Jack et al., 2016*; *Halabi et al., 2009*), even distant sites displayed a substantial divergence limit (~40%) at long evolutionary distances. This result suggests that the spatial constraints required for specific molecular function usually propagate throughout the entire protein structure and significantly limit the long-term divergence even at sites distant from catalytic residues.

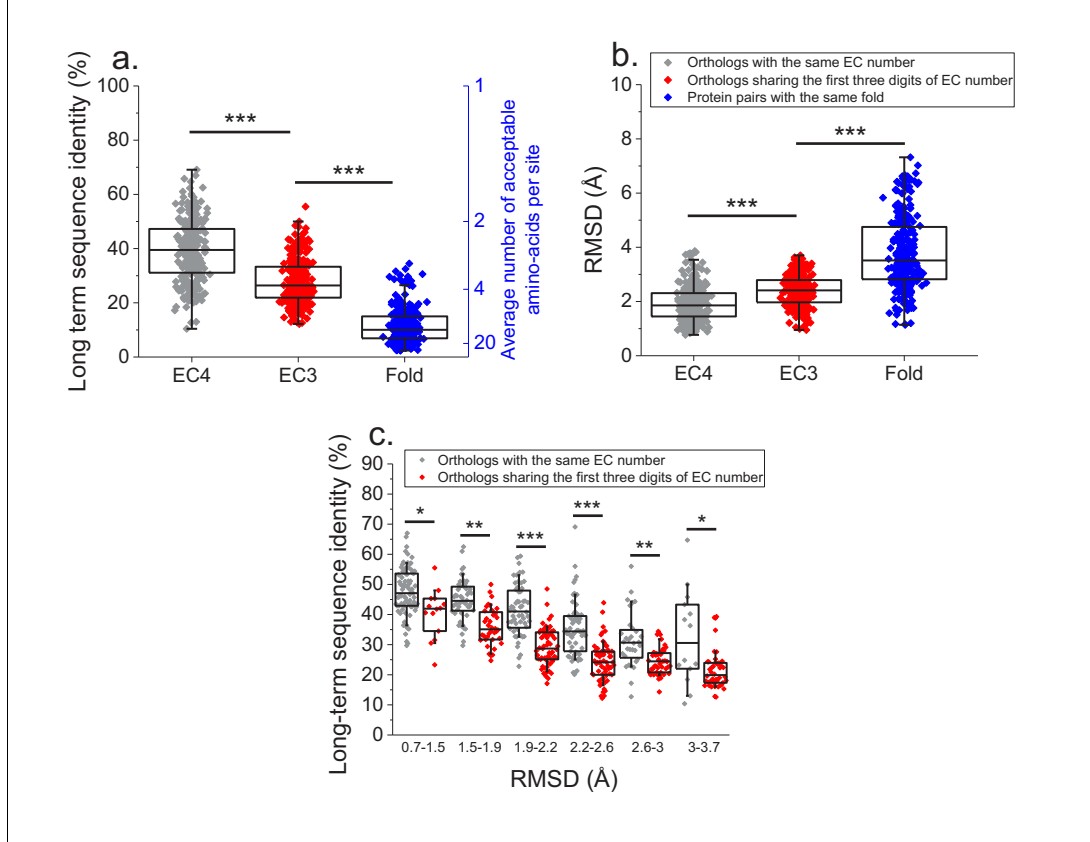

**Figure 6.** Effect of functional specificity on long-term sequence and structural divergence between orthologs. (a) Sequence identities between orthologous pairs of enzymes that diverged over two billion years ago. The long-term sequence divergence between pairs of orthologs sharing the same EC number (gray, n = 272) is compared to the divergence between pairs only sharing the first three digits of their EC numbers (red, n = 265), that is, enzymes conserving only a general class of substrates or cofactors. The results are based on enzyme COGs for the 22 species used to analyze global sequence divergence (*Supplementary file 1*). Blue points show sequence identities between pairs of proteins with the same structural fold (*Dawson et al., 2017*) but unrelated enzyme activities, that is, activities sharing no digits in the EC classification (n = 298, see Materials and methods). The right blue Y axis represents the average number of amino acid types accepted per protein site during long-term protein evolution. (b) Similar to panel a, but showing the corresponding C-alpha structural divergence (RMSD) between protein pairs. (c) Long-term sequence identities between orthologous enzyme pairs at the same levels of structural similarity. Results are shown for pairs of enzymes sharing their full EC classification (gray dots), or only sharing the first three digits of their EC classification (red dots). In all panels: *(p<0.05), **(p<1e-4), ***(p<1e-10) for the Mann-Whitney test.

DOI: https://doi.org/10.7554/eLife.39705.022

The following source data is available for figure 6:

**Source data 1.** Sequence identities between ancient orthologs sharing the same EC number and only the first three digits of their EC numbers.

DOI: https://doi.org/10.7554/eLife.39705.023

Finally, we investigated how various degrees of functional conservation affect the long-term divergence between orthologs. To that end, we compared the long-term sequence and structural similarities of enzymes sharing their full EC classification to those sharing only the first three digits of their EC classification (*Figure 6a and b*); for this comparison we only used orthologs from species with divergence times > 2 billion years (see Materials and methods). In contrast to enzymes sharing all four EC digits, conservation of the first three digits indicates only a general class of substrates or cofactors (*Bairoch, 1999*). This analysis revealed significantly lower long-term sequence identities (27% vs. 37% identity, Mann-Whitney p-value<$10^{-20}$) and structural similarities (2.4 vs. 1.8 Å RMSD, P-value $2 \times 10^{-18}$) between orthologs sharing only partial EC numbers. Notably, orthologs sharing only the first three EC digits are still substantially more conserved, both in sequence and structure

(p-values<$10^{-20}$), than pairs of enzymes with the same structural fold, but unrelated enzyme activities (i.e. enzymes sharing no digits in the EC classification) (*Figure 6a and b*, *Dawson et al., 2017*).

We also investigated the sequence constraints at the same level of protein structural divergence for proteins with different degrees of functional conservation. To that end, we calculated the sequence identity between orthologs, sharing either their full or partial EC numbers, at different bins of long-term structural similarity (*Figure 6c*). Interestingly, we observed that even at the same level of C-alpha RMSD divergence, orthologs sharing full EC numbers usually have higher levels of sequence identity compared to orthologous pairs with the same level of structural divergence but sharing only three EC digits. This result indicates that conservation of molecular function constrains sequence divergence even beyond the requirement to maintain a specific spatial structure.

## Discussion

Our analysis demonstrates that, in contrast to proteins with the same fold (*Rost, 1997*), the requirement to strictly conserve the same molecular function significantly limits the long-term sequence and structural divergence of protein orthologs. Although we confirmed the result by *Povolotskaya and Kondrashov (2010)* that ancient protein orthologs are still diverging from each other, our study reveals that the rate of this divergence becomes increasingly slow for orthologs that strictly conserve their function. Even a slight relaxation of functional specificity, for example from full to partial EC conservation (*Figure 6a and b*), leads to substantially more pronounced long-term sequence and structural divergence. Similarly, a substantial sequence identity between homologous restriction endonucleases is usually limited to isoschizomers, that is, proteins specific to the same target DNA sequence (*Pingoud et al., 2014*).

We believe that the observed divergence patterns can be explained by the following mechanistic model. Proteins with the same molecular function usually conserve the identity of their chemical and biological substrates and interaction partners. This conservation leads to functional pressure to closely preserve the spatial positions and dynamics of key protein residues necessary for efficient catalysis and function (*Lehninger et al., 2013*). In turn, the requirement to continuously preserve structural properties and functional dynamics of key protein residues likely imposes a strict requirement to preserve the overall protein structure, in other words, structural optimality necessary for efficient and specific protein function. The evolutionary pressure to preserve structural optimality (to within <3 Å C-alpha RMSD) required for a given molecular function leads, in agreement with the results by *Chothia and Lesk (1986)* and others (*Gilson et al., 2017*), to substantial levels of overall sequence conservation and the observed divergence limit. Our analysis further demonstrates that less than four amino acid types are accepted, on average, per site for proteins strictly conserving their molecular function.

We note that the observed conservation may reflect both the impact of amino acid substitutions on protein activity, due to changes in equilibrium positions and dynamics of protein residues, and on protein abundance, due to changes in overall protein stability (*Bershtein et al., 2015*; *Adkar et al., 2017*). Nevertheless, direct and comprehensive biochemical experiments demonstrated that the deleterious effects of mutations primarily arise from changes in specific protein activity rather than decreases in protein stability and cellular abundance (*Firnberg et al., 2016*). Our results support this model, demonstrating that conservation of functional specificity imposes substantially more stringent long-term sequence constraints than simply conservation of protein folds and protein stability. Indeed, while every folded protein should maintain its stability, our results demonstrate that maintaining stable folds does not constrain long-term divergence to more than 10-15% sequence identity, which is substantially less than the requirement for maintaining specific molecular functions (~40% sequence identity).

The presented results demonstrate that only about a third of the sequence conservation between distant orthologs with the same molecular function can be attributed to universally conserved protein sites, that is, sites occupied by identical amino acids in almost all lineages. In contrast, we found that different protein sites are usually identical between orthologs from different lineages. This result is likely due, at least in part, to the epistatic nature of protein sequence landscapes, where mutations that are neutral in one lineage are often prohibitively deleterious in another (*Breen et al., 2012*; *Lunzer et al., 2010*). In the context of the aforementioned divergence

model, the evolution of mitochondrial ribosomal proteins in eukaryotes (*Figure 1—figure supplement 6*) provides an interesting example, suggesting that orthologs' divergence can be substantially accelerated by co-evolution with their interaction partners or relaxation of selection pressures.

Our experimental and computational analyses also delineate two distinct stages of the long-term divergence of orthologs with the same molecular function. During the first 1–2 billion years of divergence, substitutions at protein sites with mild fitness effects lead to a substantial (40–60%) decrease in sequence identity. After the first stage, divergence at these sites effectively saturates. The saturation at sites with small fitness effects, combined with very slow divergence rate at sites with large fitness effects (*Figure 4*), leads to a substantially slower sequence and structural divergence during the second stage. Interestingly, as a consequence of this slowdown, for the past billion years there has not been a substantial decrease in sequence and structural similarity between ancient orthologs with the same molecular function. Further analyses of biochemical, biophysical and cellular constraints will reveal how various structural and functional properties influence proteins' long-term evolution, and how protein functional efficiency may be compromised by deleterious mutations (*Shendure and Akey, 2015*).

# Materials and methods

**Key resources table**

| Reagent type (species) or resource | Designation | Source or reference | Identifiers | Additional information |
|---|---|---|---|---|
| Strain, strain background (*Escherichia coli* EcNR2) | MG1655, bla, bio-, λ-Red+, mutS-::cmR | PMID: 19633652 | Addgene #26931 | |
| Sequence-based reagent | 90 bp DNA oligos with phosphorothioated bases | This paper | See *Supplementary file 4* | 100 nmole DNA Plate oligo, Integrated DNA Technologies |
| Commercial assay or kit | Miseq Reagent Kit V2 | Illumina | MS-102–2002 | |
| Commercial assay or kit | sybr green | ThermoFisher | S7567 | |
| Commercial assay or kit | Qubit HS DNA kit | ThermoFisher | Q32854 | |
| Commercial assay or kit | Q5 Hot Start High-Fidelity Mastermix | NEB | M0494S | |
| Commercial assay or kit | DNA clean and concentration kit 5 | Zymo Research | D4013 | |
| Commercial assay or kit | illustra bacteria genomicPrep Mini Spin kit | GE life sciences | 28904259 | |
| Commercial assay or kit | Agilent DNA 1000 kit | Agilent Genomics | 5067–1504 | |
| Software, algorithm | SeqPrep v1.1 | John St. John | https://github.com/jstjohn/SeqPrep | |
| Software, algorithm | Bowtie2 | PMID: 22388286 | | |
| Software, algorithm | Perl scripts to count mutant reads | This paper | https://github.com/platyias/count-MAGE-seq (copy archived at https://github.com/elifesciences-publications/count-MAGE-seq). | |

*Continued on next page*

*Continued*

| Reagent type (species) or resource | Designation | Source or reference | Identifiers | Additional information |
|---|---|---|---|---|
| Other | Turbidostat for growth competition assay | PMID: 23429717 | | |

## Considered enzyme activities and corresponding protein orthologs

We selected for analysis the sequences annotated in UniProt (*UniProt Consortium, 2015*) with EC numbers associated with the following metabolic pathways (defined in the KEGG database) (*Kanehisa et al., 2016*): Glycolysis and gluconeogenesis, pentose phosphate pathway, TCA cycle, purine metabolism, pyrimidine metabolism. Using the protein sequences from 22 diverse organisms (*Supplementary file 1*) we constructed clusters of orthologous groups (COGs) using the EdgeSearch algorithm (*Kristensen et al., 2010*). Following previous studies, we considered any two proteins from different species in the same COG as orthologs (*Tatusov et al., 1997*). COGs were obtained using the COGsoft software (*Kristensen et al., 2010*), starting from an all-against-all psi-blast (*Altschul et al., 1997*) search, setting the database size at $10^8$, and using a maximum considered E-value of 0.1. To obtain the largest number of likely orthologs we did not apply a filter on low complexity or composition-based statistics. Only proteins sharing the same EC number and assigned to the same COG were compared, and only COGs with sequences in 10 or more of the 22 species were used.

In order to exclude proteins clearly showing evidence of Horizontal Gene Transfer (HGT), we constructed a maximum likelihood phylogenetic tree of the 12 prokaryotes considered in our analysis using a concatenated alignment of marker genes (*Wu and Scott, 2012*). The species tree was then manually compared to the individual trees of the prokaryotic sequences sharing the same molecular function within each COG; COG-specific trees were built using the GAMMA model of amino-acid substitution implemented in the RAxML software (*Stamatakis, 2014*). Proteins that showed clear differences in tree topologies, suggesting HGT, were excluded from further analysis. Ancient gene duplications, that is, duplications occurring prior to the divergence between considered species, often lead to cases in which enzymes in the same COG but from different species have diverged for longer than the corresponding species' divergence times; thus, we did not consider COGs with tree topologies showing evidence of ancient gene duplications. Ancient gene duplications were defined as those occurring prior to the last common ancestor of 3 or more of the 22 species considered in the analysis.

The same procedure was used to select non-enzymatic COGs for analyses (*Figure 1—figure supplement 5*). However, in this case we only considered COGs for which none of the proteins were annotated in UniProt with metabolic EC numbers. Naturally, UniProt functional annotations for non-enzymes vary in terms of their source and format. Therefore, it is difficult to ascertain the degree of functional specificity and conservation between non-enzymatic orthologs. To address this, we manually checked that the molecular functions associated with proteins in the same COG were related, although we could not ascertain perfect conservation of molecular function.

## Models of long-term protein sequence evolution

Global sequence identities for pairs of proteins annotated with the same molecular function in the same COG were calculated using pairwise alignments with ClustalW2 (*Larkin et al., 2007*). Sequence identity was computed as the number of identical sites at aligned positions, divided by the total number of aligned sites, excluding gaps. Divergence times between organisms were obtained from the TimeTree database (*Hedges et al., 2006*) (November, 2015) and used as a proxy for protein divergence times; in the analysis we used the mean divergence times across studies listed in the database. Divergence times between bacteria and archaea were set to 4 billion years based on current estimates for the occurrence time of their Last Common Ancestor (*Sheridan et al., 2003*; *Battistuzzi et al., 2004*) and existing evidence of an early origin of life on Earth (*Bell et al., 2015*). It is likely that ancient eukaryotic genes originated through episodic endosymbiotic gene transfer events and vertical inheritance from bacterial and archaeal genomes (*Ku et al., 2015*;

*Thiergart et al., 2012*). Because of the discrete nature of such transfer events, the vast majority of individual prokaryotic-eukaryotic orthologous pairs are likely to have diverged from each other long before the origin of eukaryotes (1.8 billion years ago [*Parfrey et al., 2011*]); specifically, because most ancient prokaryotic species would not have transferred genes to eukaryotes. Thus, based on the median divergence time between the considered prokaryotes (~4 billion years, *Supplementary file 1*), divergence times between eukaryotes and prokaryotes were set in our analyses at 4 billion years. The results presented in the paper remain insensitive to the exact value of this divergence estimate (within the 3–4 billion year interval). Based on the recently proposed affiliation of eukaryotes and members from the Lokiarchaeota (*Spang et al., 2015*), divergence times between *S. solfataricus* and eukaryotes were set at 2.7 billion years, that is, the estimated age of the TACK superphylum (*Guy and Ettema, 2011*; *Betts et al., 2018*).

In order to study the long-term divergence patterns of orthologs, we only used COGs containing pairs of orthologs with at least five different divergence times distributed across 4 billion years. Sequence divergence data were fitted with models 1 to 3 using the least-squares minimization algorithm implemented in the MATLAB R2017a fitnlm function (The MathWorks, Inc, Natick, MA). The best fits of model 1 and model two were compared using the F-test. To test whether the conservation of molecular function limits protein sequence divergence, the minimum sequence identity parameter in model 2 ($Y_0$, from *Equation 2*) was compared, for each enzymatic activity, to the average global sequence identity between unrelated protein pairs using the Wald test.

To investigate the effect of the uncertainty of divergence times' estimates, we repeated the analysis of the 64 enzymatic activities while randomly assigning either the maximum or minimum value of the divergence times between lineages reported in the TimeTree database. This analysis was performed for a total of 1000 independent assignment runs. Across the independent assignment runs, the expected long-term sequence identity between orthologs was higher than 25% for at least 90% of enzymes (based on model 2), and the projected sequence identity after 7.8 billion years was above 25% (based on model 3) for at least 75% of enzymes (*Figure 1—figure supplement 7*).

To assess the effect of computational functional annotations on the observed divergence results, we repeated the analysis using only sequences with experimentally validated molecular functions (*Figure 1—figure supplement 3*). To keep only sequences with validated molecular functions, we manually reviewed published references for enzyme annotations in the BRENDA database (*Chang et al., 2015*), and discarded any functional assignments that were based exclusively on computational or high-throughput studies. After filtering for the experimentally validated annotations, we only considered EC numbers corresponding to pairs of orthologs with at least four different divergence times distributed across 4 billion years.

## Calculation of the divergence rate

Based on Model 3, we determined the divergence rate, that is, the rate of the decrease in sequence identity per time, at a given divergence time *t* by solving for the derivative of *Equation 3* with respect to time:

$$\frac{dy}{dt} = \frac{d\left(100 * \left(\frac{R_0 * t}{\alpha} + 1\right)^{-\alpha}\right)}{dt} = -100 * R_0 \left(\frac{R_0 * t}{\alpha} + 1\right)^{-\alpha - 1} \tag{5}$$

where *y* represents global sequence identity, *t* represents divergence time, $R_0$ represents the average substitution rate, and α represents the shape parameter of the gamma distribution.

## Equivalency between model two and a poisson divergence model with allowed back substitutions

In the Jukes-Cantor model of nucleotide divergence (*Tajima and Nei, 1984*; *Jukes and Cantor, 1969*), the expected number of substitutions per site (δ) between two sequences after a divergence time *t* from a common ancestor is given by:

$$\delta = -\frac{a-1}{a} ln\left(1 - \frac{a}{a-1}(1-y)\right) \tag{6}$$

where *y* is the proportion of identical sites and *a* is the number of allowed nucleotide types (usually 4). The same model can be applied to the divergence of protein sequences (*Yang et al., 2000*;

*Ota and Nei, 1994*), by setting *a* to the number of allowed amino acid types per protein site. Furthermore, $\delta = 2\lambda t$, where $\lambda$ represents the substitution rate per site per unit time, which is assumed to be equal across all sites. Substituting $\delta$, and solving the above equation for *y* yields:

$$y = \frac{1}{a} + \left(1 - \frac{1}{a}\right)\exp\left(-\frac{2\lambda a}{a-1}t\right) \tag{7}$$

which is mathematically equivalent to model 2 *Equation 2*, with $R_0 = \frac{2\lambda a}{a-1}$, and $Y_0 = \frac{1}{a}$. Thus, $Y_0$ can also be interpreted as the inverse of the average number of amino acids accepted per protein site during protein evolution.

## FolA competition experiment in *E. coli*

To perform competition experiments we used the EcNR2 strain derived from *E. coli* K12 MG1655. Mutagenesis was performed using Multiplex Automated Genomic Engineering (MAGE), as previously described (*Wang et al., 2009*). 90 bp DNA oligomers were designed around each folA codon using the MG1655 wild type sequence as reference (*Supplementary file 4*). For each codon, all possible nucleotide variants were synthesized. To avoid simultaneous mutations of multiple codons, cells were transformed targeting ten consecutive codons at a time. After four rounds of electroporation, cells were recovered and pooled together at approximately the same concentration based on cell counts. Two competition growth experiments were carried out, one for each half of the protein. For the competition experiments, cells were grown in LB media in a turbidostat while maintaining constant volume and cell density. Samples were taken every 2 hr for a period of 16 hr, spun down, washed in PBS, spun down again and stored at $-20°C$ until all samples were collected. For each competition, the corresponding FolA region was amplified through PCR while assigning a specific DNA barcode for each time point. PCR products were then pooled and paired-end sequenced using the MiSeq Reagent Kit two from Illumina. Sequence reads were deposited to the SRA database with accession number: SRP152339.

To determine, at each time point, the abundance of each mutant relative to wild type, we joined paired-end reads using SeqPrep (v 1.1) and aligned the joined reads to the folA gene sequence using Bowtie2 (*Langmead and Salzberg, 2012*). We then counted the number of reads per mutant using a custom script (*Plata, 2018*). Reads with more than a single mutated codon were discarded. Counts were median-normalized to control for noise due to mutagenesis performed in batches of 10 codons. At each time point we calculated the ratio $R_t$ of mutant to wild type (WT) reads. In exponential growth, the growth rate difference between a given mutant and WT was calculated based on the slope of $\ln(R_\tau)$ as a function of time:

$$\ln(R_t) = (m_i - m_{wt}) * t + \ln(R_0)$$

where $m_i$ and $m_{wt}$ represent the mutant and WT growth rates, respectively. Growth rate differences were calculated only for mutants with at least five time points with 20 or more reads. Relative growth rates were calculated by dividing the slopes in the equation above by the number of e-fold increases given the average dilution rate of the turbidostat (1.37/h).

To calculate a single value characterizing the effect of all possible mutations at a protein site, we first averaged the relative growth rates of mutants resulting in the same amino acid change. We then calculated the average fitness effect of mutations at each protein site by averaging across 20 possible amino acids substitutions (*Supplementary file 3*).

To estimate the sensitivity of our results to sequencing errors, we calculated the average fitness effect of substitutions at each FolA site using the relative growth rates of mutant strains carrying only 32 mutated codons selected at random out of 64 possible codons. We observed a high correlation (Pearson's r: 0.95, p-value<1e-20, *Figure 4—figure supplement 2*) between the average growth rate effects at each site calculated using two non-overlapping subsets of 32 codons. As expected, nonsense mutations and substitutions in the *folA* start codon had substantially stronger average effects on growth rates compared to other substitutions (26% versus 4% slower growth than WT, respectively. Mann Whitney U, p-value<$10^{-20}$). Also, the relative growth rates due to synonymous codon substitutions were usually very mild (0.2% higher growth compared to WT); 97% of synonymous substitutions had growth effects of less than 3%.

## Contribution of different sites to the divergence limit

In order to identify phylogenetically independent pairs of species, we aligned the 16S rRNA gene sequences of bacterial species having orthologs annotated with the target 30 EC numbers (*Figure 3—figure supplement 1*). 16S rRNA sequences were obtained from the GreenGenes database (*DeSantis et al., 2006*) (October, 2016). We then built maximum likelihood phylogenetic trees based on the 16S alignments using RAxML (*Stamatakis, 2014*). Next, we used the Maximum Pairing Problem approach by *Arnold and Stadler (2010)* to find the largest number of edge-disjoint pairs of species with 16S rRNA genetic distances corresponding to >2 billion years of divergence. Divergence times were estimated from the 16S genetic distances based on the linear regression of literature reported divergence times (*Hedges et al., 2006*) (*Supplementary file 1*). The F84 model of nucleotide substitution implemented in the phylip package (*Felsenstein, 2005*) was used to compute the genetic distances. Using the 16S alignment data, we calculated the probability that a protein site was identical across independent lineages. The probability was calculated as the fraction of orthologous pairs from phylogenetically independent species pairs with identical amino acids at the site. The amino acid identities at a given site were obtained based on the multiple sequence alignment of all orthologs associated with each EC number, obtained using ClustalW2 (*Larkin et al., 2007*). A similar procedure was applied to analyze FolA and InfA orthologs from the COG database (*Figure 3c*).

To investigate the divergence of sites with different fitness effects, we used sequences of FolA and InfA bacterial orthologs from the COG database (*Galperin et al., 2015*). The FolA orthologs annotated with the same EC number in UniProt (n = 311) and the InfA orthologs annotated with the same KEGG Orthology (KO) number in KEGG (n = 514) were used to build multiple sequence alignments with ClustalW2 (*Larkin et al., 2007*). Divergence times were estimated from the 16S genetic distances as described above. Within each divergence bin (*Figure 4a,b*), sequence identities of sites with different average fitness effects (represented by different colors in *Figure 4a,b*) were averaged across all pairs of orthologs at a given divergence time.

## Analysis of global protein structural evolution

To study the divergence of protein structures as a function of time, we obtained PDB codes for all proteins associated with EC numbers in the BRENDA database (*Chang et al., 2015*). We then selected for the analysis species with experimentally solved enzyme structures for at least 10 different EC numbers. Psi-blast searches with a conservative E-value cutoff of $10^{-6}$ were used to identify orthologs (defined as bi-directional best hits) in the selected species. The 3D structures of orthologous pairs, annotated with the same EC number, were aligned using the TM-align program (*Zhang and Skolnick, 2005*) to obtain the C-alpha RMSD values. Pairs of proteins were not considered if more than 70% of the residues of the shortest protein could not be structurally aligned. We also removed from the analysis pairs of structures with flexibility between domains, as they could result in large RMSD values despite significant structural similarity. To identify such proteins we used the FATCAT (*Ye and Godzik, 2003*) software to perform flexible structural alignments of all structure pairs. We then filtered the structural pairs that were split into two or more domains by the FATCAT alignments.

## Analysis of the enzyme active sites

To analyze divergence as a function of active site distance we used protein sequences associated with the 64 EC numbers and 22 species considered in *Figure 1* and *Figure 1—figure supplement 1*. To that end, PDB (*Berman et al., 2000*) was searched for homologous sequences annotated with the same enzymatic activities and with known 3D structures. Annotations of active site residues for the corresponding structures were obtained from the Catalytic Site Atlas (*Porter, 2004*), PDB and UniProt-KB (*UniProt Consortium, 2015*). For each PDB structure with available active site information, protein sites were then stratified into different layers according to the distance between their alpha carbons and the centroid of the active site residues. Each pair of orthologs was then aligned using ClustalW2 (*Larkin et al., 2007*) with a homolog in PDB annotated with the same activity and with defined distance layers around the active site; the PDB sequence with the highest sequence identity to either member of the pair was used for the alignment. Sequence identities for different

layers were calculated based on the structural positions in the corresponding PDB reference sequences.

### Comparison of pairs of enzymes with the same structural folds

We used structural classifications of protein domains from the CATH database (v4.2.0) (*Dawson et al., 2017*). For structural comparisons, we only considered PDB structures with a single classified domain per chain. Protein pairs classified in CATH in the same homologous structural superfamily were considered as having the same fold. To obtain pairs of proteins in the same fold but with different functions, we only considered PDB structures annotated with different EC numbers in BRENDA. For this analysis we randomly selected 300 pairs of structures with the same fold that do not share any digits of their EC classification.

## Acknowledgements

We sincerely thank Dan Tawfik, Eugene Koonin, and Fyodor Kondrashov for very helpful discussions. This work was supported in part by the National Institute of General Medical Sciences grants GM079759 and R35GM13188 to DV.

## Additional information

### Funding

| Funder | Grant reference number | Author |
| --- | --- | --- |
| National Institute of General Medical Sciences | R01 | Dennis Vitkup |

The funders had no role in study design, data collection and interpretation, or the decision to submit the work for publication.

### Author contributions

Mariam M Konaté, Data curation, Software, Formal analysis, Investigation, Methodology, Writing—original draft; Germán Plata, Data curation, Software, Formal analysis, Investigation, Methodology, Writing—original draft, Writing—review and editing; Jimin Park, Investigation, Methodology, Writing—review and editing; Dinara R Usmanova, Formal analysis, Investigation, Methodology, Writing—review and editing; Harris Wang, Supervision, Methodology, Project administration, Writing—review and editing; Dennis Vitkup, Conceptualization, Supervision, Funding acquisition, Methodology, Writing—original draft, Project administration, Writing—review and editing

### Author ORCIDs

Germán Plata https://orcid.org/0000-0002-6470-7748
Dinara R Usmanova https://orcid.org/0000-0001-5031-0013
Harris Wang http://orcid.org/0000-0003-2164-4318
Dennis Vitkup https://orcid.org/0000-0003-4259-8162

### Decision letter and Author response

Decision letter https://doi.org/10.7554/eLife.39705.034
Author response https://doi.org/10.7554/eLife.39705.035

## Additional files

### Supplementary files

• Supplementary file 1. Considered model species and pairwise average divergence times.
DOI: https://doi.org/10.7554/eLife.39705.024

• Supplementary file 2. Fitted model parameters, statistical test results and non-enzymatic protein annotations. *Supplementary file 2A* Fitted model parameters and test results for the 64 considered

activities (EC numbers). *Supplementary file 2B* Estimated rates of sequence divergence for pairs of orthologs according to Model three fits *Supplementary file 2C* Fitted model parameters and test results for 29 sets of orthologs not annotated with EC numbers *Supplementary file 2D* UniProt annotations of representative sequences from E. coli and H. sapiens for sets of orthologs not annotated with EC numbers *Supplementary file 2E* UniProt annotations of representative sequences from E. coli and H. sapiens for sets of orthologs not annotated with EC numbers and fast divergence rates in eukaryotes.

DOI: https://doi.org/10.7554/eLife.39705.025

• Supplementary file 3. Average relative growth rate effect of amino acid substitutions in the FolA protein of *E. coli*.

DOI: https://doi.org/10.7554/eLife.39705.026

• Supplementary file 4. DNA oligomers used to introduce amino acid substitutions along the FolA protein.

DOI: https://doi.org/10.7554/eLife.39705.027

• Transparent reporting form

DOI: https://doi.org/10.7554/eLife.39705.028

### Data availability

Sequence reads were deposited to the SRA database with accession number: SRP152339. Source data files have been provided for figure 1, figure 5, figure 6 and figure 1 supplements 1, 3, 5 and 6.

The following dataset was generated:

| Author(s) | Year | Dataset title | Dataset URL | Database and Identifier |
|---|---|---|---|---|
| Konaté MM, Plata G, Park J, Usmanova DR, Wang HH, Vitkup D | 2018 | Molecular function limits divergent protein evolution on planetary timescales | https://www.ncbi.nlm.nih.gov/sra/SRP152339 | SRA, SRP152339 |

The following previously published dataset was used:

| Author(s) | Year | Dataset title | Dataset URL | Database and Identifier |
|---|---|---|---|---|
| Eric D Kelsic, Hattie Chung, Niv Cohen, Jimin Park, Harris H Wang, Roy Kishony | 2016 | Data S1 | https://www.cell.com/cms/attachment/2118262336/2085801250/mmc5.zip | Supplementary material of: https://doi.org/10.1016/j.cels.2016.11.004, 10.1016/j.cels.2016.11.004 |

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
