## [Decision Letter]

[**Editorial note:** This article has been through an editorial process in which the authors decide how to respond to the issues raised during peer review. The Reviewing Editor's assessment is that all the issues have been addressed.]

Thank you for submitting your article "Molecular function limits divergent protein evolution on planetary timescales" for consideration by *eLife*. Your article has been reviewed by three peer reviewers, including Nir Ben-Tal as the Reviewing Editor and Reviewer #1, and the evaluation has been overseen by Diethard Tautz as the Senior Editor. The other reviewers remain anonymous.

The Reviewing Editor has highlighted the concerns that require revision and/or responses, and we have included the separate reviews below for your consideration. If you have any questions, please do not hesitate to contact us.

Summary:

The manuscript analyzes clusters of orthologue enzymes of the same function, defined as sharing the same 4-digits EC number, to examine the limitation of function preservation on evolutionary divergence. The results seem to indicate that function confines divergence, which is not surprising. The potential novelty here is the time dependence of the process, where phylogenetic time of species divergence is used as natural time scale "clock". The data shows that proteins diverse from each other rapidly during the first ~1.5 billion years, but saturate after that.

Opinion: The manuscript addresses an important and interesting topic, but it should be revised significantly to clarify what exactly was computed and why, what was measured and why, and how the results are related to previous studies. In view of the amount and nature of the issues raised, the authors should consider withdrawing the manuscript.

Major concerns:

(pretty much copy-pasted from the individual reports)

1) This study poses the question about the extent to which function constrains sequence divergence. The answer given is that for orthologous enzymes 25% sequence identity is the approximate limit. The major critique, is that the authors did not make an attempt to provide a clear mechanistic explanation for such phenomenon. Is it simply because a significant fraction of positions is invariant? Is it because there are various constraints on various sites? Is it because only a small subsets of amino acids are tolerated at some sites? Why not take several enzyme families with very thick alignments and analyze them position-by-position to shed light on this question? Without such deep analysis, the paper, while interesting in its general idea, reads superficial and underdeveloped, despite some experimental results.

2) The authors premise their study on the assertion made by Rost in a 1997 review that proteins of the same fold can diverge up to random sequence ID of 3-5%, i.e., that structural similarity does not impose constraint on sequence ID. As a consequence the authors attribute in their analysis all sequence identity constraints to functional similarity. However, this premise is highly problematic. While anecdotally one can find two proteins of similar fold with very low sequence ID, typically proteins with similar structure (not necessarily orthologues) have sequence ID at or above 25% – this is the essence of the famous cusp in sequence-structure relationship first reported by Lesk and Chothia in the eighties. In fact a careful analysis of structural/stability constraints on sequence divergence has been published in recent paper (Biophys J v.112, p.1350-65, 2017) where it was shown that in divergent evolution scenario protein structure is maintained up to 25-30% sequence ID and quickly deteriorates beyond that. Incidentally it is very close to 25% ID which the authors claim to stem from functional constraints. Furthermore, the above mentioned BJ paper presents a detailed analytical estimate of sequence divergence dynamics akin to exponential fits used in this PAPER. The authors should make themselves familiar with this theory and consider using it to fit their divergence curves rather than ad hoc exponential fits.

3) The high throughput experimental data is potentially interesting but its description is so cryptic and incomplete that it is very hard to assess what actually has been done there. In particular there is no assessment of noise in deep sequencing to assess fitness, of the effects of synonymous substitutions, etc. How is the data binned and why binning? The use of LB as a medium for the folA experiment is unfortunate because in LB DHFR is much less essential than in minimal media and therefore the results could be skewed by very permissive conditions. In addition the standing genetic variation is not taken into account – fitness effects could be a result of hitchhiking (i.e. originate from the variation of the background into which the mutations are introduced). These concerns aside, it is not entirely clear what does comparison with experiment tell us beyond the obvious that sites where deleterious fitness effects are greatest evolve more slowly. Was the sole purpose of the experiment to bin positions by their fitness effects? If so, what new insight is gained from slower divergence of sites with more profound fitness effects compared to faster divergence of sites that are more tolerant to substitutions in experiments. Isn't this effect expected? Maybe quantification of the effect may have some novelty, but such quantification has not been done. The obvious thing to do is to compare their alleged fitness effects with dN/dS assessment for the rates but that has not been done.

4) The word "protein" should be replaced with "enzyme" in the title. "We focused on enzymes because their molecular function is usually well defined," doesn't sound quite true. Reading the manuscript, it seems that the goal of this work is to investigate divergence of enzymes. The divergence of non-enzymes will most likely follow a different pattern and will generally be much lower than 25% identity. If the authors want to generalize to non-enzymes, they need to perform appropriate analyses. Currently, only one experiment is dedicated to a non-enzyme. It is not fair to the readers to generalize beyond the scope of analysis that was carried out.

5) The sentence in the Abstract "divergence rates of orthologous enzymes decrease substantially after ~1-2 billion years of independent evolution" is confusing. "Rate" means number of amino acid substitutions per site per unit of time. Moreover, the authors write in the Introduction "a divergence limit does not imply that the rate of amino acid substitutions slows down in evolution," thus contradicting the Abstract. The Abstract should be edited and the misleading sentence corrected. From the context of the paper is seems like by "divergence rate" the author mean the rate of decrease in pairwise sequence identity with time. This definition poses additional question. This "divergence rate" will always decrease due to multiple and back substitutions and it is trivial. The authors need to make it clear that what they mean goes beyond a trivial curvilinear relationship between time and sequence identity.

6) Further on this, evolutionary rate is much more than simply a count of changes in the identity of the amino acids. To measure it properly one needs to rely on the phylogeny. Indeed, since the study is based on several clusters of orthologous proteins, why not do just that? For example, why not use the dN/dS formalism?

7) "billion years of evolution, we observed a significant decrease in mutual divergence rates," see item 5 above. Also, shouldn't "significant" be quantified is some way?

8) "Species' divergence times were used to estimate the times of divergence between corresponding orthologous proteins": How reliable is this time estimate?

9) Discussion "The presented results demonstrate that, in contrast to proteins with the same fold [Rost, 1997], the requirement to maintain the same molecular function significantly constrains the long-term divergence of protein orthologs": This difference is surprising. Maybe it has to do with how changes were estimated in the two studies? To state it with confidence direct comparison is needed, where the exact same methodology is used.

10) Finally, the manuscript should be edited for clarity by a professional scientific editor.

Separate reviews (please respond to each point):

*Reviewer #1:*

Summary of paper:

The manuscript analyzes clusters of orthologue enzymes of the same function, defined as sharing the same 4-digits EC number, to examine the limitation of function preservation on evolutionary divergence. The results seem to indicate that function confines divergence, which is anticipated- perhaps also shown before. The potential novelty here is the time dependence of the process. The data shows (to the best of my understanding) that proteins diverse from each other rapidly during the first ~1.5 billion years, but saturate after that. The data further suggests (again, assuming that I understood correctly) that the rapid evolutionary phase mostly reflects random drift, i.e., amino acid sites that can freely mutate, and that the saturation value reflects their relative fraction of the entire enzyme sequence.

Opinion: The manuscript could be important but it should be revised significantly to clarify what exactly was computed and why, what was measured and why, and how the results are related to previous studies. I also believe the manuscript should be edited for clarity by a professional scientific editor.

Major issues:

1) Results "In the first model, all protein sites have equal and independent substitution rates; see Equation 1": Equation 1 obviously represents an unrealistic evolutionary model because it is well known that the evolutionary rates are not equal at all amino acid sites. For example, catalytic residues hardly change.

2) Where does Equation 2 come from? It also doesn't look particularly realistic.

3) Equation 3 is the commonly used evolutionary model I think. So why not start with it?

4) "Species' divergence times were used to estimate the times of divergence

between corresponding orthologous proteins": How reliable is this time estimate?

5) Figure 2: How is "fitness cost" defined?

6) "phylogenetically independent pairs of species": What does it mean and how is "independent" defined? Is the pair A-B in Figure 3A phylogenetically independent? And the pair D-H?

7) "Notably, the probability that a protein site is identical, and thus contributes to the divergence limit, first increases linearly with increasing fitness effects at the site, and then begins to saturate for sites with high (>30%) fitness effects (Figure 3B)." This is perhaps true for FolA but I do not see saturation for InfA.

8) The distributions of the probability of identical sites in FolA and InfA (Figures 4A and B) are very different, but they are discussed together without reference to the difference.

9) Discussion "The presented results demonstrate that, in contrast to proteins with the same fold [Rost, 1997], the requirement to maintain the same molecular function significantly constrains the long-term divergence of protein orthologs": I am surprised about this difference. Maybe it has to do with how changes were estimated in the two studies? To state it with confidence I would have liked to see direct comparison where the exact same methodology is used.

10) Evolutionary rate is much more than simply a count of changes in the identity of the amino acids. To measure it properly one needs to rely on the phylogeny. Indeed, since the study is based on several clusters of orthologous proteins, why not do just that?

*Reviewer #2:*

This study poses the question about the extent to which function constrains sequence divergence. The answer given is that for orthologous enzymes 25% sequence identity is the approximate limit. The major critique, is that the authors did not make an attempt to provide a clear mechanistic explanation for such phenomenon. Is it simply because a significant fraction of positions is invariant? Is it because there are various constraints on various sites? Is it because only a small subsets of amino acids are tolerated at some sites? Why not take several enzyme families with very thick alignments and analyze them position-by-position to shed light on this question? Without such deep analysis, the paper, while interesting in its general idea, reads superficial and underdeveloped, despite some experimental results.

1) I suggest replacing the word "protein" with "enzyme" in the title. "We focused on enzymes because their molecular function is usually well defined," doesn't sound quite true. Reading the manuscript, it seems that the goal of this work is to investigate divergence of enzymes. I am positive that the divergence of non-enzymes will follow a different pattern and will generally be much lower than 25% identity. If the authors want to generalize to non-enzymes, they need to perform appropriate analyses. Currently, only one experiment is dedicated to a non-enzyme. It is not fair to the readers to generalize beyond the scope of analysis that was carried out.

2) I think that the sentence in the Abstract "divergence rates of orthologous enzymes decrease substantially after ~1-2 billion years of independent evolution" is confusing. "Rate" usually means number of amino acid substitutions per site per unit of time. Moreover, the authors write in the Introduction "a divergence limit does not imply that the rate of amino acid substitutions slows down in evolution," thus contradicting the Abstract. I suggest to edit the Abstract and correct the misleading sentence. From the context of the paper is seems like by "divergence rate" the author mean the rate of decrease in pairwise sequence identity with time. This definition poses additional question. This "divergence rate" will always decrease due to multiple and back substitutions and it is trivial. The authors need to make it clear that what they mean goes beyond a +trivial curvilinear relationship between time and sequence identity.

3) "billion years of evolution, we observed a significant decrease in mutual divergence rates," see item 2 above. Also, shouldn't "significant" be quantified is some way?

4) It was not made clear why these experiments were performed and how they integrate with the rest of the study. Addition of these experiments seems preliminary and no confident conclusions follow. Was the sole purpose of the experiment to bin positions by their fitness effects? If so, what new insight is gained from slower divergence of sites with more profound fitness effects compared to faster divergence of sites that are more tolerant to substitutions in experiments. Isn't this effect expected? Maybe quantification of the effect may have some novelty, but such quantification has not been done.

*Reviewer #3:*

In this paper the authors aim to assess how does the requirement of conserved function (i.e. orthology) constraints sequence evolution. The author premise their study on the assertion made by Rost in 1997 review that proteins of the same fold can diverge up to random sequence ID of 3-5% i.e. that structural similarity does not impose constraint on sequence ID. As a consequence the authors attribute in their analysis all sequence identity constraints to functional similarity. To that end they analyze sequence divergence of orthologs using phylogenetic time of species divergence as natural time scale "clock". They observed saturating time dependencies of sequence divergence with time and conclude that the rate of sequence divergence decreases with decreasing divergence time. Further, they carried out a high throughput mutational experiment on fitness effects of substitutions in 2 genes – folA and InfA – in *E. coli* and find, perhaps unsurprisingly, that sites where mutational effects on fitness are stronger are less constrained in evolution.

While the paper contains some interesting bioinformatics observations and analysis I have serious concerns about its premise and interpretation of the results. Apparently some issues stem from author's apparent gaps in knowledge of modern literature on biophysical determinants of protein evolution.

1) The premise that structural similarity does not constraint sequence identity is highly problematic. While anecdotally one can find two proteins of similar fold with very small sequence ID typically proteins with similar structure (not necessarily orthologues) have sequence ID at or above 25% – this is the essence of the famous cusp in sequence-structure relationship first reported by Lesk and Chothia in the eighties. In fact a careful analysis of structural/stability constraints on sequence divergence has been published in recent paper (Biophys J v.112, p.1350-65 (2017) where it was shown that in *divergent* evolution scenario protein structure is maintained up to 25-30% sequence ID and quickly deteriorates beyond that. Incidentally it is very close to 25% ID which the authors claim to stem from functional constraints Furthermore, the above mentioned BJ paper presents a detailed analytical estimate of sequence divergence dynamics akin to exponential fits used in this PAPER. The authors should make themselves familiar with this theory and use it to for their divergence curves rather ad hoc exponential fits.

2) The high throughput experimental data is potentially interesting but its description is so cryptic and incomplete that it is very hard to assess what actually has been done there. In particular there is no assessment of noise in deep sequencing to assess fitness., of the effects of synonymous substitutions etc. How is the data binned and why binning? The use of LB as a medium for the folA experiment is unfortunate because in LB DHFR is much less essential than in minimal media and therefore the results could be skewed by very permissive conditions. In addition the standing genetic variation is not taken into account – fitness effects could be a result of hitchhiking (i.e. originate from the variation of the background into which the mutations are introduced).These concerns aside, it is not entirely clear what does comparison with experiment tell us beyond the obvious that sites where deleterious fitness effects are greatest evolve more slowly. The obvious thing to do is to compare their alleged fitness effects with dN/dS assessment of the rates but that has not been done.

3) The interpretation of the experimental fitness effects in terms of function is also questionable. The authors are apparently unaware of series of experimental works from Shakhnovich lab where determinants of fitness effects of mutations are addressed. In particular it has been shown using both point mutations and orthologous chromosomal replacements for folA gene (PLOS Genetics 2015 DOI:10.1371/journal.pgen.1005612) and adk gene (Nature Ecology Evolution, 2017 http://dx.doi.org/10.1038/s41559-017-0149) that fitness is determined by product of folded protein abundance A and activity kcat/KM. Mutations may affect stability and through that parameter A (by changing the balance between protein production and degradation, see Mol Cell v.49, pp133-44 (2013). Therefore interpretation of the experimental trends entirely in functional terms is not warranted.

A minor comment: The concept and metaphor of expanding protein universe has been introduced 10 years before Kondrashov's work in the paper "Expanding protein universe and its origin from biological big bang" PNAS 2002, v.99 pp. 14132-6.

[Editors' note: further revisions were suggested before publication, as described below.]

Thank you for resubmitting your work entitled "Molecular function limits divergent protein evolution on planetary timescales" for further consideration at *eLife*. Your revised article has been favorably evaluated by Diethard Tautz as the Senior Editor, a Reviewing Editor, and two reviewers.

As you can see from the reports below, the reviewers appreciated the revisions. However, there are still major outstanding issues. While some of these can be resolved by changes in the presentation, others are fundamental. We would strongly encourage you to address all of these prior to publication.

*Reviewer #2:*

I find that the authors did a thorough revision of the manuscript. At least now I think I understood the main conclusion of the paper. In enzymes and other proteins with very strong functional conservation, the number of different amino acids acceptable at a position is about 4. It is not because many sites are invariant and some are variable (not a dirty trick of average temperature in a hospital), but because most sites (except the invariant ones) are constrained to use a library of 3 to 5 different amino acids, not more than that. If this is not the bottom line, then the authors need to do better job at crystallizing their main claim and result.

If it is, I think it is a meaningful finding that could be explained better to the readers. I guess the second claim is that 3-5 amino acid limit is universal to all enzymes and (conserved!) non-enzymes. I do doubt (as in the original review) the validity of such a strong claim, which could be a result of the authors' bias in selecting families for their analysis. At least for non-enzymes, they selected most conserved proteins (like ribosomal proteins), so of course such selection is biased to get proteins that saturate easily in evolution. The authors try to justify this biased selection suggesting that it is difficult to find orthologs. But that statement by itself totally discredits this study. Why? Because if you cannot find your orthologs, wouldn't it mean that they already diverged beyond (lower than) your claimed 25% identity as the universal limit, and the author's conclusions do not apply to such families? Maybe for the enzymes too, the 25% limit is simply a reflection of the search methods the authors used to find orthologs, that fail to find more distant ones.

To improve the presentation and make this paper quite interesting (well, reviewers are not supposed to direct the study, but the authors seem passionate about their work and also seem rather inexperienced in both logical thinking and putting a paper together, so maybe this recommendation could be helpful), I would suggest to base it on two plots.

1) Between-protein variability.

The first plot is the histogram of the average estimated number of amino acids allowable per site (without invariant sites, or with) for enzyme families and other protein families. Well, the authors would have to try harder to find orthologs for proteins that manage to diverge below 25%, even my rotation students can do that. This histogram would be expected to have a maximum around 3 to 5, and for some variable proteins it could be around 10 or 15? Right? Then protein families with very low and very high numbers could be discussed with an attempt to give explanations about their uniqueness.

2) Within-protein variability.

The second plot is the histogram of estimated number of amino acids allowable per site within a protein family. The authors could either normalize to the average per protein, or select a bin with the maximum count from previous plot (let's say 4). These histograms can be averaged for all families with mean and SD showing for each bin. I would assume there will be a large count for invariant sites for enzymes (at 1), counts for 2 amino acids used, 3, 4, etc. Will this histogram have a single mode? Maybe around 4? Several modes? Will there be sites using more than 10 amino acids? How are these distributed in spatial structure and relative to the active site? Discussion of these details could be quite insightful and interesting.

If these authors do not wish to make these plots, since this review will be published, maybe someone else will, and we will learn something interesting as a result.

Currently, there are so many plots in this paper and many of them are not particularly helpful to get the point across quickly. Also I still find the usage of non-standard terms (like divergence rate) confusing, not necessary, and not insightful about the mechanism. Yes, the terms are defined, but they are just masking the reality, which is simpler: constrained usage of amino acids in positions of enzyme molecules. Not all 20, not even 10, but 4! Why not state and illustrate this clearly? Don't you agree that the impact of the paper will increase because it will be easier to understand?

The Abstract is still very poor and misleading. For instance, the statement "The effective divergence limit (>25% sequence identity) is not primarily due to multiple substitutions at the same sites" is completely wrong. According to the authors' explanations, this "effective divergence limit" is exactly due to multiple substitutions at the same site! If you have only 4 amino acids acceptable at a site, due to multiple substitutions sequence identity will saturate are 25%. Like in DNA. Why not write a precise and clear Abstract about what this work presents? Due to so many logical flaws in the authors' thinking and presentation (as illustrated above), I would be scared to publish this paper without a careful read. One statement at a time. And trying to figure out why the statement is wrong. If clearly not wrong, then move on to the next one.

And, finally, why not compare your results with this paper more thoroughly PMID: 27138088? Isn't it a bit similar? I guess I don't understand the meaning behind "to investigate the temporal patterns of the long-term divergence."

Do the authors still claim that the sites are saturated at the usage of 3 to 5 amino acids because the time passed was not sufficient to gain more changes? Or the time was enough and protein of the same function simply cannot tolerate additional amino acids well and still keep the function? Which one is right? I got an impression it was the latter. And then what I said at the beginning of this review holds. If it is the former, it needs to be convincingly justified.

If the authors disagree with my review, then I did not understand the paper, which is possible, and still suggests that the authors need to improve the presentation.

*Reviewer #3:*

This version of the manuscript is a significant improvement over the previous version in terms of added details (e.g. experimental procedure of folA mutagenesis re now described in sufficient detail to be understandable and/or potentially reproducible).

The authors attempted to address my (and other reviewers) main concern by presenting the analysis in new Figure 5 that shows that full orthologues (all EC numbers coincide) are more sequence-constrained than partial orthologues (3 EC numbers coincide). However this analysis fails to control for different structural divergence between full and partial orthologues.

Therefore, the most problematic aspect of the analysis – that authors attribute observed sequence conservation in diverging clades to conservation of function between orthologues has not been fully addressed. A clear alternative explanation that such conservation is explained by maintenance of structure and stability regardless of function still stands.

The authors used a truism by pointing out that proteins of the same FOLD (i.e. topologically similar arrangement of elements of secondary structure) can diverge to 10% or less sequence ID again citing an old work with data collected on very limited number of structures available at that time.

However, their full functional orthologues are much more similar structurally than proteins sharing the same fold. The correct control which was suggested in my initial review has not been done satisfactorily. Specifically, the authors should compare their sets of proteins with sets that have similar degree of structural similarity (measured as distribution of TM-scores) but different function. There are such examples where structures are quite similar (TM-score-wise) but functions differ significantly. Good examples of that kind ate TIM-barrels which are almost exclusively enzymes with wide variety of specificity and Igfold proteins again with very conserved structures but broadly diverged functional annotations.

If the authors find that conservation of structure in functionally divergent proteins imposes less sequence divergence constraints than same degree of structural conservation in orthologues – that will be a clear demonstration of additional constraints imposed by functional conservation which is the main message of this work. In the absence of such analysis the current data does not justify the conclusion.

Minor point: The authors severely misquote Firnberg et al., 2016. They say: "Nevertheless, direct and comprehensive biochemical experiments demonstrated that the deleterious effects of protein mutations primarily result from changes in specific protein activity rather than decreases in protein stability and cellular abundance [Firnberg et al., 2016]". In fact, the authors of Firnberg et al., 2016, say directly the opposite: '… These DFEs provide insight into the inherent benefits of the genetic code's architecture, support for the hypothesis that mRNA stability dictates codon usage at the beginning of genes, an extensive framework for understanding protein mutational tolerance, and evidence that mutational effects on protein thermodynamic stability shape the DFE..…" (cited from the Abstract of Firnberg et al., 2016). The authors misrepresent the main result of Firnberg et al., 2016: According to Figure 5B of Firnberg et al., 2016, the *product* of abundance and catalytic activity shapes fitness effects in TEM1, not abundance alone. This is exactly what is established in Bershtein et al., 2015 and Adkar et al., 2017, and shows that abundance (which is a function of stability) enters fitness landscape on equal footing with catalytic activity, i.e. there is as much selection for abundance (i.e. stability) as it is for kcat/KM and related measures of activity. This misinterpretation somewhat undermines the major premise of the present paper that there is separate selection for activity/function and stability/structure. In reality one cannot disentangle the two because fitness landscape depends on the function (i.e. the product) of the two factors.

---

## [Author Response]

The manuscript analyzes clusters of orthologue enzymes of the same function, defined as sharing the same 4-digits EC number, to examine the limitation of function preservation on evolutionary divergence. The results seem to indicate that function confines divergence, which is not surprising. The potential novelty here is the time dependence of the process, where phylogenetic time of species divergence is used as natural time scale "clock". The data shows that proteins diverse from each other rapidly during the first ~1.5 billion years, but saturate after that.Opinion: The manuscript addresses an important and interesting topic, but it should be revised significantly to clarify what exactly was computed and why, what was measured and why, and how the results are related to previous studies. In view of the amount and nature of the issues raised, the authors should consider withdrawing the manuscript.

We sincerely thank the reviewing editor and reviewers for their thoughtful suggestions and comments. Following the reviewers’ suggestions, we have substantially revised the manuscript to clarify the main results, their novelty and importance. Although the fact that functional conservation constrains protein divergence is indeed not surprising, our manuscript describes interesting long-term temporal patterns and the limits of this divergence for proteins with the same molecular function. We welcome the opportunity provided by this innovative peer review model to share with the readers our replies to the reviewers’ comments. We believe that this scientific dialogue will be an important contribution to understanding of long-term evolution of protein sequences and structures.

Major concerns:(pretty much copy-pasted from the individual reports)1) This study poses the question about the extent to which function constrains sequence divergence. The answer given is that for orthologous enzymes 25% sequence identity is the approximate limit. The major critique, is that the authors did not make an attempt to provide a clear mechanistic explanation for such phenomenon. Is it simply because a significant fraction of positions is invariant? Is it because there are various constraints on various sites? Is it because only a small subsets of amino acids are tolerated at some sites? Why not take several enzyme families with very thick alignments and analyze them position-by-position to shed light on this question? Without such deep analysis, the paper, while interesting in its general idea, reads superficial and underdeveloped, despite some experimental results.

We apologize for the possible confusion. The mechanistic explanation of our results is the following. We demonstrate in the manuscript that the long-term evolution of proteins that continuously maintain the same molecular functions generally requires a strict conservation of protein structures (<3Å C-α root mean square deviation). The structural conservation is likely necessary for proper structural positioning of active site residues required for efficient function and catalysis. This level of structural conservation, in agreement with the Chothia and Lesk analysis (Chothia and Lesk, 1986), requires a substantial conservation of protein sequences, which leads to the observed sequence divergence limit. Furthermore, our analysis shows that only about 35% of the sequence conservation between distant orthologs can be attributed to universally conserved protein sites, i.e. sites occupied by identical amino acids in almost all lineages. In contrast, the observed divergence limit primarily originates from a small number of acceptable amino acids at each protein site under the continuous constraint to strictly conserve protein structure and function. Following the reviewers’ suggestion, we now present an analysis of sequence conservation across protein sites using multiple sequence alignments for 30 diverse enzyme families (Figure 2B, Figure 2—figure supplement 1).

2) The authors premise their study on the assertion made by Rost in a 1997 review that proteins of the same fold can diverge up to random sequence ID of 3-5%, i.e., that structural similarity does not impose constraint on sequence ID. As a consequence the authors attribute in their analysis all sequence identity constraints to functional similarity. However, this premise is highly problematic. While anecdotally one can find two proteins of similar fold with very low sequence ID, typically proteins with similar structure (not necessarily orthologues) have sequence ID at or above 25% – this is the essence of the famous cusp in sequence-structure relationship first reported by Lesk and Chothia in the eighties. In fact a careful analysis of structural/stability constraints on sequence divergence has been published in recent paper (Biophys J v.112, p.1350-65, 2017) where it was shown that in divergent evolution scenario protein structure is maintained up to 25-30% sequence ID and quickly deteriorates beyond that. Incidentally it is very close to 25% ID which the authors claim to stem from functional constraints. Furthermore, the above mentioned BJ paper presents a detailed analytical estimate of sequence divergence dynamics akin to exponential fits used in this PAPER. The authors should make themselves familiar with this theory and consider using it to fit their divergence curves rather than ad hoc exponential fits.

We believe that there may be a potential misunderstanding here in terms of the definition of protein folds and structures, and in terms of the nature of evolutionary constraints. A protein fold is defined as a particular set of protein secondary structures, their arrangement and topological connections (Murzin et al., 1995). Both the Rost 1997 paper and the Biophysical J. paper (cited by the reviewer) actually show that a low sequence identity (<25%) between proteins with the same fold is very common. Thus, the sequence constraints necessary to simply maintain a protein fold cannot be responsible for the divergence limit we observe. We also would like to emphasize that selection does not specifically act to maintain structural similarity; instead, structural and sequence conservation is a consequence of the selection for a molecular function necessary to maintain species fitness. In contrast to protein folds, our analysis demonstrates that the continuous conservation of specific molecular functions usually requires high structural similarity (<3Å C-α root mean square deviation), and that the requirement for the structural conservation likely constrains the divergence of sequence identity (>25%).

In terms of the analytical model presented in the Biophysical J. 2017 paper, we note that the exponential formulas used in our fits are not *ad hoc*. They directly correspond to a divergence model with equal probability of mutations across protein sites, as in the Biophysical J. model. The BJ model is numerically identical to our second model. Specifically, the fraction of identical sites (*y)* after a certain divergence time (*t)* in the BJ model is given by:

y=1-((a-1)/a)(1-(1-a/l(a-1)^μt^)

according to Equation 7 in the BJ paper, where the parameter *l* is the length of the protein, *a* is the number of acceptable amino acid types per protein site, and *μ* is the per-protein substitution rate. After re-arranging, and given that the substitution rate per site *μ_0_* =*μ × l^-1^*:

y=1a+(1-1/a)(1+((-a/(a-1))/l))^lμ_0_t^

Given the well-known limit: exp(x)=lim_n→∞_⁡(1+x/n)^n^, and because protein length *l* is substantially larger than a/(a-1):

y=1/a+(1-1/a)exp(-R_0_t); with R0=(a/(a-1))μ0

This equation is numerically identical to our model 2. Furthermore, this demonstrates that the parameter *Y_0_* in model 2 can be also interpreted as the inverse of the effective number of amino acid types acceptable per protein site when molecular function is strictly conserved. Our results thus suggest that, on average, only 2 to 4 amino acids types are usually accepted per protein site as long as protein molecular function is conserved.

3) The high throughput experimental data is potentially interesting but its description is so cryptic and incomplete that it is very hard to assess what actually has been done there. In particular there is no assessment of noise in deep sequencing to assess fitness, of the effects of synonymous substitutions, etc. How is the data binned and why binning? The use of LB as a medium for the folA experiment is unfortunate because in LB DHFR is much less essential than in minimal media and therefore the results could be skewed by very permissive conditions. In addition the standing genetic variation is not taken into account – fitness effects could be a result of hitchhiking (i.e. originate from the variation of the background into which the mutations are introduced). These concerns aside, it is not entirely clear what does comparison with experiment tell us beyond the obvious that sites where deleterious fitness effects are greatest evolve more slowly. Was the sole purpose of the experiment to bin positions by their fitness effects? If so, what new insight is gained from slower divergence of sites with more profound fitness effects compared to faster divergence of sites that are more tolerant to substitutions in experiments. Isn't this effect expected? Maybe quantification of the effect may have some novelty, but such quantification has not been done. The obvious thing to do is to compare their alleged fitness effects with dN/dS assessment for the rates but that has not been done.

We agree with the reviewers that the correlation between the fitness effects of mutations and evolutionary conservation is indeed expected. But this particular result was neither a goal nor a major conclusion of our analysis. Our primary goals were to characterize the long-term temporal divergence patterns at sites with different fitness effects, to explore the limits of the sequence divergence at different sites, and to investigate the contribution of sites with various fitness effects to the divergence limit. For example, our analysis demonstrated (Figure 3), for the first time to our knowledge, that the divergence at sites with relatively small fitness effects usually saturates after 1-2 billion years of evolution, while the divergence at sites with higher fitness effects still continues, although at a substantially smaller rate.

In our analysis, we binned the sites (based on their average fitness effects) in order to explore the sequence divergence for sites within each fitness category. We used logarithmic bins in our analysis due to the skewed distribution of fitness effects in the experimental data (Figure 3—figure supplement 2).

Following the reviewers’ suggestion, we now use several approaches to demonstrate the reliability of our experimental measurements: 1.) we show that mutants targeting start and stop codons have, on average, significantly larger growth effects compared to amino acid replacements; 2.) we show that synonymous substitutions yield very small fitness effects; and 3.) by calculating average growth defects per site (based on strains with non-overlapping sets of introduced codons), we demonstrate a high reproducibility (r = 0.95) of our experimental findings. Overall, these results demonstrate that neither sequencing noise nor background mutations have a significant effect on the experimental measurements.

4) The word "protein" should be replaced with "enzyme" in the title. "We focused on enzymes because their molecular function is usually well defined," doesn't sound quite true. Reading the manuscript, it seems that the goal of this work is to investigate divergence of enzymes. The divergence of non-enzymes will most likely follow a different pattern and will generally be much lower than 25% identity. If the authors want to generalize to non-enzymes, they need to perform appropriate analyses. Currently, only one experiment is dedicated to a non-enzyme. It is not fair to the readers to generalize beyond the scope of analysis that was carried out.

We indeed believe that for enzymes, specific molecular functions are usually well-defined. Following the reviewers’ suggestion, we now also present the results of similar sequence analyses for orthologous proteins that are not enzymes. In order to explore the long-term divergence limit, protein orthologs need to be present across very long evolutionary distances. Therefore, the analyzed proteins mostly represent ribosomal proteins, chaperones and electron transport flavoproteins. Notably, the results obtained for these protein families are consistent with the divergence limit observed for enzymes (Figure 1—figure supplement 5). We also now describe an interesting and specific case of several mitochondrial ribosomal proteins which display much faster divergence rates compared to their prokaryote orthologs; a likely consequence of their co-evolution with the fast-evolving mitochondrial-encoded rRNA.

5) The sentence in the Abstract "divergence rates of orthologous enzymes decrease substantially after ~1-2 billion years of independent evolution" is confusing. "Rate" means number of amino acid substitutions per site per unit of time. Moreover, the authors write in the Introduction "a divergence limit does not imply that the rate of amino acid substitutions slows down in evolution," thus contradicting the Abstract. The Abstract should be edited and the misleading sentence corrected. From the context of the paper is seems like by "divergence rate" the author mean the rate of decrease in pairwise sequence identity with time. This definition poses additional question. This "divergence rate" will always decrease due to multiple and back substitutions and it is trivial. The authors need to make it clear that what they mean goes beyond a trivial curvilinear relationship between time and sequence identity.

The reviewer is correct, and we apologize for being unclear. We specifically define the sequence divergence rate (now in the third paragraph of the Introduction) as the decrease in the pairwise sequence identity per unit time. Notably, this divergence rate is different from the substitution rate, which corresponds to the number of amino acid substitutions per site per unit time. We have also edited the Abstract and the main text to further clarify this distinction. As we discuss in the paper (fifth paragraph of the Results) the observed divergence limit is not a trivial consequence of the curvilinear relationship between divergence time and sequence identity. For example, according to the data fits using model 1 and model 3, the decrease in the sequence divergence rate after ~1-2 billion years of evolution is more than an order of magnitude higher than expected simply due to multiple substitutions at the same sites. We show in the paper that the observed sequence divergence limit is primarily due to a small number of acceptable amino acids at each protein site under the continuous and strict conservation of protein function.

6) Further on this, evolutionary rate is much more than simply a count of changes in the identity of the amino acids. To measure it properly one needs to rely on the phylogeny. Indeed, since the study is based on several clusters of orthologous proteins, why not do just that? For example, why not use the dN/dS formalism?

We would like to clarify that the primary goal of our manuscript was not to determine the short-term evolutionary rates across different sites, which can be indeed quantified using the dN/dS formalism. Our main goals were to characterize the patterns of long-term sequence divergence across protein sites and the contribution of sites to the divergence limit for proteins with the same molecular function. Unfortunately, short-term substitution rates (based on the dN/dS formalism) are not informative across long evolutionary distances (billions of years) due to saturation of substitutions at synonymous sites. We rely in our analyses on phylogenetic information by using the consensus species' divergence times (Supplementary file 1).

7) "billion years of evolution, we observed a significant decrease in mutual divergence rates," see item 5 above. Also, shouldn't "significant" be quantified is some way?

Following the reviewer’s comment, we have re-worded this sentence to clarify that we are specifically referring to the decline of sequence identity per unit time; we also now report that, based on model 3 and Equation 5, the sequence divergence rate for proteins with the same molecular function decreases more than ten times after ~1.5 billion years of evolution.

8) "Species' divergence times were used to estimate the times of divergence between corresponding orthologous proteins": How reliable is this time estimate?

We thank the reviewer for this question. We now present in Supplementary file 1 data describing the variance in the estimated divergence times based on multiple independent publications. Overall, most estimates of species divergence differ by tens to several hundred million years. To assess the sensitivity of our results to this variation, we repeated the analysis using ~1000 independent runs by randomly sampling either the maximum or minimum reported divergence times between each pair of lineages. The fits of the divergence model 2 based these simulations demonstrate that for ~90% of proteins the independent assignments of the divergence times support a long-term sequence identity limit higher than 25% (Figure 1—figure supplement 7).

9) Discussion "The presented results demonstrate that, in contrast to proteins with the same fold [Rost, 1997], the requirement to maintain the same molecular function significantly constrains the long-term divergence of protein orthologs": This difference is surprising. Maybe it has to do with how changes were estimated in the two studies? To state it with confidence direct comparison is needed, where the exact same methodology is used.

We agree with the reviewer that this result is both interesting and surprising. This is one of the key results of our manuscript. This result is especially interesting given that proteins with the same fold often have very low levels of sequence similarity. Our analysis demonstrates that, in contrast to protein that simply share the same fold, orthologs that continuously conserve specific molecular functions usually display high degree of structural and sequence similarity even after billions of years of divergent evolution. We also show (Figure 5) that even a small relaxation of functional conservation (for example, from sharing all 4 digits of the EC number to only sharing the first 3 digits of the EC number) results in a significant increase of long-term sequence divergence.

10) Finally, the manuscript should be edited for clarity by a professional scientific editor.

Following the reviewer’s suggestion, we have extensively revised the text to improve clarity, and to highlight the novelty and importance of our findings.

Separate reviews (please respond to each point):

Reviewer #1:

Summary of paper:The manuscript analyzes clusters of orthologue enzymes of the same function, defined as sharing the same 4-digits EC number, to examine the limitation of function preservation on evolutionary divergence. The results seem to indicate that function confines divergence, which is anticipated- perhaps also shown before. The potential novelty here is the time dependence of the process. The data shows (to the best of my understanding) that proteins diverse from each other rapidly during the first ~1.5 billion years, but saturate after that. The data further suggests (again, assuming that I understood correctly) that the rapid evolutionary phase mostly reflects random drift, i.e., amino acid sites that can freely mutate, and that the saturation value reflects their relative fraction of the entire enzyme sequence.Opinion: The manuscript could be important but it should be revised significantly to clarify what exactly was computed and why, what was measured and why, and how the results are related to previous studies. I also believe the manuscript should be edited for clarity by a professional scientific editor.Major issues:1) Results "In the first model, all protein sites have equal and independent substitution rates; see Equation 1": Equation 1 obviously represents an unrealistic evolutionary model because it is well known that the evolutionary rates are not equal at all amino acid sites. For example, catalytic residues hardly change.2) Where does Equation 2 come from? It also doesn't look particularly realistic.3) Equation 3 is the commonly used evolutionary model I think. So why not start with it?

All scientific models are ultimately approximations to reality. They are essentially useful tools for answering specific questions and testing hypotheses. For example, despite being clearly an approximation to reality, model 2 fits the long-term sequence divergence data pretty well (Figure 1). Notably, increasing the complexity of the models (e.g. model 2 compared to model 3) does not significantly improve the fits to the data (see R^2^ values in Supplementary file 2A). Model 2 describes sequence divergence with independent and equal probability of amino acid substitutions across protein sites and a divergence limit at long evolutionary distances.

Our main reason for first considering models 1 and 2 is to test whether the observed decline in protein sequence identity (i.e. the sequence divergence rate) is simply due to multiple substitutions at the same sites (a trivial explanation for the curvilinear relationship between time and sequence identity), or whether it is necessary to assume additional constraints, such as the long-term divergence limit. Testing the goodness of fits between these two simple models allows us to answer this specific question. We now describe in the text that model 2 represents a simple modification to model 1 in which the exponential decay of sequence identity is bounded by a minimum identity at long evolutionary distances.

4) "Species' divergence times were used to estimate the times of divergencebetween corresponding orthologous proteins": How reliable is this time estimate?

We thank the reviewer for this question. We now present in Supplementary file 1 data describing the variance in the estimated divergence times based on multiple independent publications. Overall, most estimates of species divergence differ by tens to several hundred million years. To assess the sensitivity of our results to this variation, we repeated the analysis using ~1000 independent runs by randomly sampling either the maximum or minimum reported divergence times between each pair of lineages. The fits of the divergence model 2 based these simulations demonstrate that for ~90% of proteins the independent assignments of the divergence times support a long-term sequence identity limit higher than 25% (Figure 1—figure supplement 7).

5) Figure 2: How is "fitness cost" defined?

Fitness cost was defined as the percent decrease in bacterial growth rate relative to the original strain, i.e. the “wild type”, in which mutations were introduced. We have now clarified this definition in the manuscript.

6) "phylogenetically independent pairs of species": What does it mean and how is "independent" defined? Is the pair A-B in Figure 3A phylogenetically independent? And the pair D-H?

Phylogenetically independent pairs of species are defined as pairs of species that do not share any edges in the phylogenetic tree with each other since each pair diverged from a common ancestor. In Figure 2A of the updated manuscript, the pair A-B is independent from the pair D-H because they do not share edges in the phylogenetic tree since divergence from a common ancestor. By contrast, the pair D-H is not independent from the pair E-F. We now provide these examples in the main text to clarify these points.

7) "Notably, the probability that a protein site is identical, and thus contributes to the divergence limit, first increases linearly with increasing fitness effects at the site, and then begins to saturate for sites with high (>30%) fitness effects (Figure 3B)." This is perhaps true for FolA but I do not see saturation for InfA.

We agree with the reviewer that the saturation is more prominent for FolA. However, we also observe it in InfA; for example, for InfA there is a larger (>2-fold) change in the probability of site identity in the 10-40% interval of the relative growth rate decrease (x-axis), compared to the 40-70% interval.

8) The distributions of the probability of identical sites in FolA and InfA (Figures 4A and B) are very different, but they are discussed together without reference to the difference.

We apologize for being unclear. Likely due to family-specific constraints, the distributions of the probability of identical sites are indeed quite different between FolA and InfA. But our main point in this analysis is that in both cases the fraction of universally conserved sites, i.e. sites conserved in almost all lineages, is relatively small. We now present in Figure 2B and Figure 2—figure supplement 1 a more comprehensive analysis based on 30 different enzyme families. These results demonstrate that sites conserved in the majority of the linages are responsible for only about a third of the overall sequence identity between distant orthologs.

9) Discussion "The presented results demonstrate that, in contrast to proteins with the same fold [1997], the requirement to maintain the same molecular function significantly constrains the long-term divergence of protein orthologs": I am surprised about this difference. Maybe it has to do with how changes were estimated in the two studies? To state it with confidence I would have liked to see direct comparison where the exact same methodology is used.

We agree with the reviewer that this result is interesting and surprising. This is one of the key results of our manuscript. Previous studies demonstrated (for example, Rost, 1997), that proteins sharing the same fold often have very small degrees of sequence and structural similarity. Our analysis shows that, in contrast to proteins that simply share the same fold, orthologs that continuously conserve specific molecular functions usually display high degree of structural and sequence similarity even after billions of years of divergent evolution. We also show (Figure 5) that even a small relaxation of functional conservation (for example, from sharing all 4 digits of the EC number to only sharing the first 3 digits of the EC number) results in a significant increase of long-term sequence divergence.

10) Evolutionary rate is much more than simply a count of changes in the identity of the amino acids. To measure it properly one needs to rely on the phylogeny. Indeed, since the study is based on several clusters of orthologous proteins, why not do just that?

We agree with the reviewer that evolutionary rate is more than a count of amino acid changes, which is why we use multiple divergence models to fit the data. We note that we indeed use phylogenetic information by considering published divergence times between different lineages. However, we do not fit an evolutionary model to the clusters of orthologous proteins because the short-term evolutionary rates derived in this way would not be informative about divergence patterns over long geological timescales. Our study is focused on understanding the patterns of long-term divergence of sequence and structural identity for proteins conserving molecular function, rather than estimating short-term evolutionary rates across protein sites.

Reviewer #2:

This study poses the question about the extent to which function constrains sequence divergence. The answer given is that for orthologous enzymes 25% sequence identity is the approximate limit. The major critique, is that the authors did not make an attempt to provide a clear mechanistic explanation for such phenomenon. Is it simply because a significant fraction of positions is invariant? Is it because there are various constraints on various sites? Is it because only a small subsets of amino acids are tolerated at some sites? Why not take several enzyme families with very thick alignments and analyze them position-by-position to shed light on this question? Without such deep analysis, the paper, while interesting in its general idea, reads superficial and underdeveloped, despite some experimental results.

We thank the reviewer for suggestions and apologize for the potential confusion. The mechanistic explanation of our results is the following. We demonstrate in the manuscript that the long-term evolution of proteins that continuously maintain the same molecular functions generally requires a strict conservation of protein structures (<3Å C-α root mean square deviation). The structural conservation is likely necessary for proper structural positioning of active site residues required for efficient function and catalysis. This level of structural conservation, in agreement with the Chothia and Lesk analysis (Chothia and Lesk, 1986), requires a substantial conservation of protein sequences, which leads to the observed sequence divergence limit. Furthermore, our analysis shows that only about 35% of the sequence conservation between distant orthologs can be attributed to universally conserved protein sites, i.e. sites occupied by identical amino acids in almost all lineages. In contrast, the observed divergence limit primarily originates from a small number of acceptable amino acids at each protein site under the continuous constraint to strictly conserve protein structure and function. Following the reviewers’ suggestion, we now present an analysis of sequence conservation across protein sites using multiple sequence alignments for 30 diverse enzyme families (Figure 2B, Figure 2—figure supplement 1).

1) I suggest replacing the word "protein" with "enzyme" in the title. "We focused on enzymes because their molecular function is usually well defined," doesn't sound quite true. Reading the manuscript, it seems that the goal of this work is to investigate divergence of enzymes. I am positive that the divergence of non-enzymes will follow a different pattern and will generally be much lower than 25% identity. If the authors want to generalize to non-enzymes, they need to perform appropriate analyses. Currently, only one experiment is dedicated to a non-enzyme. It is not fair to the readers to generalize beyond the scope of analysis that was carried out.

We indeed believe that for enzymes, specific molecular functions are usually well-defined. Following the reviewers’ suggestion, we now also present the results of similar sequence analyses for orthologous proteins that are not enzymes. In order to explore the long-term divergence limit, protein orthologs need to be present across very long evolutionary distances. Therefore, the analyzed proteins mostly represent ribosomal proteins, chaperones and electron transport flavoproteins. Notably, the results obtained for these protein families are consistent with the divergence limit observed for enzymes (Figure 1—figure supplement 5). We also now describe an interesting and specific case of several mitochondrial ribosomal proteins which display much faster divergence rates compared to their prokaryote orthologs; a likely consequence of their co-evolution with the fast-evolving mitochondrial-encoded rRNA.

2) I think that the sentence in the Abstract "divergence rates of orthologous enzymes decrease substantially after ~1-2 billion years of independent evolution" is confusing. "Rate" usually means number of amino acid substitutions per site per unit of time. Moreover, the authors write in the Introduction "a divergence limit does not imply that the rate of amino acid substitutions slows down in evolution," thus contradicting the Abstract. I suggest to edit the Abstract and correct the misleading sentence. From the context of the paper is seems like by "divergence rate" the author mean the rate of decrease in pairwise sequence identity with time. This definition poses additional question. This "divergence rate" will always decrease due to multiple and back substitutions and it is trivial. The authors need to make it clear that what they mean goes beyond a +trivial curvilinear relationship between time and sequence identity.

The reviewer is correct, and we apologize for being unclear. We specifically define the sequence divergence rate (now in the third paragraph of the Introduction) as the decrease in the pairwise sequence identity per unit time. Notably, this divergence rate is different from the substitution rate, which corresponds to the number of amino acid substitutions per site per unit time. We have also edited the Abstract and the main text to further clarify this distinction. As we discuss in the paper (fifth paragraph of the Results) the observed divergence limit is not a trivial consequence of the curvilinear relationship between divergence time and sequence identity. For example, according to the data fits using model 1 and model 3, the decrease in the sequence divergence rate after ~1-2 billion years of evolution is more than an order of magnitude higher than expected simply due to multiple substitutions at the same sites. We show in the paper that the observed sequence divergence limit is primarily due to a small number of acceptable amino acids at each protein site under the continuous and strict conservation of protein function.

3) "billion years of evolution, we observed a significant decrease in mutual divergence rates," see item 2 above. Also, shouldn't "significant" be quantified is some way?

Following the reviewer’s comment, we have re-worded this sentence to clarify that we are specifically referring to the decline of sequence identity per unit time; we also now report that, based on model 3 and Equation 5, the sequence divergence rate for proteins with the same molecular function decreases about ten times after ~1.5 billion years of evolution.

4) It was not made clear why these experiments were performed and how they integrate with the rest of the study. Addition of these experiments seems preliminary and no confident conclusions follow. Was the sole purpose of the experiment to bin positions by their fitness effects? If so, what new insight is gained from slower divergence of sites with more profound fitness effects compared to faster divergence of sites that are more tolerant to substitutions in experiments. Isn't this effect expected? Maybe quantification of the effect may have some novelty, but such quantification has not been done.

We agree with the reviewer that the correlation between the fitness effects of mutations and evolutionary conservation is expected. But this particular result was neither a goal nor a major conclusion of our analysis. Our primary goals were to characterize the long-term temporal divergence patterns at sites with different fitness effects, to explore the limits of the sequence divergence across sites, and to investigate the contribution of sites with various fitness effects to the divergence limit. For example, our analysis demonstrated (Figure 3), for the first time to our knowledge, that the divergence at sites with relatively small fitness effects usually saturates after 1-2 billion years of evolution, while the divergence at sites with higher fitness effects still continues, although at a substantially smaller rate.

Reviewer #3:

[…] While the paper contains some interesting bioinformatics observations and analysis I have serious concerns about its premise and interpretation of the results. Apparently some issues stem from author's apparent gaps in knowledge of modern literature on biophysical determinants of protein evolution.1) The premise that structural similarity does not constraint sequence identity is highly problematic. While anecdotally one can find two proteins of similar fold with very small sequence ID typically proteins with similar structure (not necessarily orthologues) have sequence ID at or above 25% – this is the essence of the famous cusp in sequence-structure relationship first reported by Lesk and Chothia in the eighties. In fact a careful analysis of structural/stability constraints on sequence divergence has been published in recent paper (Biophys J v.112, p.1350-65 (2017) where it was shown that in divergent evolution scenario protein structure is maintained up to 25-30% sequence ID and quickly deteriorates beyond that. Incidentally it is very close to 25% ID which the authors claim to stem from functional constraints Furthermore, the above mentioned BJ paper presents a detailed analytical estimate of sequence divergence dynamics akin to exponential fits used in this PAPER. The authors should make themselves familiar with this theory and use it to for their divergence curves rather ad hoc exponential fits.

We believe that there may be a potential misunderstanding here in terms of the definition of protein folds and structures, and in terms of the nature of evolutionary constraints. A protein fold is defined as a particular set of protein secondary structures, their arrangement and topological connections (Murzin et al., 1995). Both the Rost 1997 paper and the Biophysical J. paper (cited by the reviewer) actually show that a low sequence identity (<25%) between proteins with the same fold is very common. Thus, the sequences constraints necessary to simply maintain a protein fold cannot be responsible for the divergence limit we observe. We also would like to emphasize that selection does not specifically act to maintain structural similarity; instead, structural and sequence conservation is a consequence of the selection for a molecular function necessary to maintain species fitness. In contrast to protein folds, our analysis demonstrates that the continuous conservation of specific molecular functions usually requires high structural similarity (<3Å C-α root mean square deviation), and that the requirement for the structural conservation likely constrains the divergence of sequence identity (>25%).

In terms of the analytical model presented in the Biophysical J. 2017 paper, we note that the exponential formulas used in our fits are not *ad hoc*. They directly correspond to a divergence model with equal probability of mutations across protein sites, as in the Biophysical J. model. The BJ model is numerically identical to our second model. Specifically, the fraction of identical sites (*y)* after a certain divergence time (*t)* in the BJ model is given by:

y=1-((a-1)/a)(1-(1-a/l(a-1))^μt^)

according to Equation 7 in the BJ paper, where the parameter *l* is the length of the protein, *a* is the number of acceptable amino acid types per protein site, and *μ* is the per-protein substitution rate. After re-arranging, and given that the substitution rate per site *μ_0_* = *μ × l^-1^*:

y=1/a+(1-1/a)(1+(-a/(a-1))/l)^lμ_0_t^

Given the well-known limit: exp(x)=lim_n→∞⁡_(1+x/n)^n^, and because protein length *l* is substantially larger than a/(a-1):

y=1/a+(1-1/a)exp(-R_0_t); with R0=(a/(a-1))μ_0_

This equation is numerically identical to our model 2. Furthermore, this demonstrates that the parameter *Y_0_* in model 2 can be also interpreted as the inverse of the effective number of amino acid types acceptable per protein site when molecular function is strictly conserved. Our results thus suggest that, on average, only 2 to 4 amino acids types are usually accepted per protein site as long as protein molecular function is conserved.

2) The high throughput experimental data is potentially interesting but its description is so cryptic and incomplete that it is very hard to assess what actually has been done there. In particular there is no assessment of noise in deep sequencing to assess fitness., of the effects of synonymous substitutions etc. How is the data binned and why binning? The use of LB as a medium for the folA experiment is unfortunate because in LB DHFR is much less essential than in minimal media and therefore the results could be skewed by very permissive conditions. In addition the standing genetic variation is not taken into account – fitness effects could be a result of hitchhiking (i.e. originate from the variation of the background into which the mutations are introduced).These concerns aside, it is not entirely clear what does comparison with experiment tell us beyond the obvious that sites where deleterious fitness effects are greatest evolve more slowly. The obvious thing to do is to compare their alleged fitness effects with dN/dS assessment of the rates but that has not been done.

We agree with the reviewers that the correlation between the fitness effects of mutations and evolutionary conservation is indeed expected. But this particular result was neither a goal nor a major conclusion of our analysis. Our primary goals were to characterize the long-term temporal divergence patterns at sites with different fitness effects, to explore the limits of the sequence divergence at different sites, and to investigate the contribution of sites with various fitness effects to the divergence limit. For example, our analysis demonstrated (Figure 3), for the first time to our knowledge, that the divergence at sites with relatively small fitness effects usually saturates after 1-2 billion years of evolution, while the divergence at sites with higher fitness effects still continues, although at a substantially smaller rate.

We would also like to clarify that the primary goal of our manuscript was not to determine the short-term evolutionary rates across different sites, which can be indeed quantified using the dN/dS formalism. Our main goals were to characterize the patterns of long-term sequence divergence for proteins with the same molecular function. Short-term substitutions rates (calculated using the dN/dS formalism) cannot reveal the patterns of sequence divergence across billions of years of evolution due to saturation of substitutions at synonymous sites.

In our analysis, we binned the sites (based on their average fitness effects) in order to explore the sequence divergence for sites within each fitness category. We used logarithmic bins in our analysis due to the skewed distribution of fitness effects in the experimental data (Figure 3—figure supplement 2).

Following the reviewers’ suggestion, we now use several approaches to demonstrate the reliability of our experimental measurements: 1.) we show that mutants targeting start and stop codons have, on average, significantly larger growth effects compared to amino acid replacements; 2.) we show that synonymous substitutions yield very small fitness effects; and 3.) by calculating average growth defects per site (based on strains with non-overlapping sets of introduced codons), we demonstrate a high reproducibility (r = 0.95) of our experimental findings. Overall, these results demonstrate that neither sequencing noise nor background mutations have a significant effect on the experimental measurements.

3) The interpretation of the experimental fitness effects in terms of function is also questionable. The authors are apparently unaware of series of experimental works from Shakhnovich lab where determinants of fitness effects of mutations are addressed. In particular it has been shown using both point mutations and orthologous chromosomal replacements for folA gene (PLOS Genetics 2015 DOI:10.1371/journal.pgen.1005612) and adk gene (Nature Ecology Evolution, 2017 http://dx.doi.org/10.1038/s41559-017-0149) that fitness is determined by product of folded protein abundance A and activity kcat/KM. Mutations may affect stability and through that parameter A (by changing the balance between protein production and degradation, see Mol Cell v.49, pp133-44 (2013). Therefore interpretation of the experimental trends entirely in functional terms is not warranted.

We are aware of the experimental work from the Shakhnovich lab, and agree with the reviewer that the effects of mutations on fitness could arise either due to effects on protein activity or protein stability, which affects the product of protein abundance and activity. We note however that direct and comprehensive biochemical experiments demonstrate that the deleterious effects of protein mutations primarily arise from changes in specific protein activity rather than decreases in stability or protein cellular abundance; see, for example, Firnberg et al., 2014. We now clarify this point in the Discussion.

A minor comment: The concept and metaphor of expanding protein universe has been introduced 10 years before Kondrashov's work in the paper "Expanding protein universe and its origin from biological big bang" PNAS 2002, v.99 pp. 14132-6.

We thank the reviewer for bringing this manuscript to our attention, and we now cite this paper in the main text.

[Editors' note: further revisions were suggested before publication, as described below.]As you can see from the reports below, the reviewers appreciated the revisions. However, there are still major outstanding issues. While some of these can be resolved by changes in the presentation, others are fundamental. We would strongly encourage you to address all of these prior to publication.

We again thank the editors and reviewers for their remarks and suggestions. We have updated the manuscript to fully address the comments and remaining concerns. Following the recommendation of the second reviewer, we highlighted the low average number of acceptable amino acids per site associated with the conservation of protein molecular function. Following the suggestions of the third reviewer, we further clarified the effects of functional constraints on the divergence limit both for protein sequences and structures. We also demonstrated, as the third reviewer requested, that even at the same level of structural divergence, orthologs sharing more specific molecular functions display substantially higher levels of long-term sequence identity.

Reviewer #2:I find that the authors did a thorough revision of the manuscript. At least now I think I understood the main conclusion of the paper. In enzymes and other proteins with very strong functional conservation, the number of different amino acids acceptable at a position is about 4. It is not because many sites are invariant and some are variable (not a dirty trick of average temperature in a hospital), but because most sites (except the invariant ones) are constrained to use a library of 3 to 5 different amino acids, not more than that. If this is not the bottom line, then the authors need to do better job at crystallizing their main claim and result.

We thank the reviewer for the comments. Indeed, the observation that only a few (less than 4) amino acids are accepted, on average, per site for proteins with conserved molecular function is one of our main results, which we now highlight in the Abstract. However, as we explained in our previous reply, this is not the only main result or conclusion of our work.

We apologize for repeating our statements from a previous reply. In the paper we asked several important and interrelated questions: 1) How far can two sequences diverge while continuously maintaining the same specific molecular function? 2) What are the temporal patterns of this divergence across billions of years of evolution? and 3) How do different protein sites contribute to the long-term divergence between orthologs with the same molecular function? Our paper provides corresponding main conclusions: 1) Orthologs that share the same molecular function usually do not diverge beyond ~3.5 Å RMSD and ~25% sequence identity (this translates to 3-4 amino acids, on average, per site). 2) The decrease of sequence identity between diverging orthologs becomes increasingly slow past 1-2 billion years of divergent evolution, such that ancient orthologs have retained approximately the same level of similarity for the past billion years. And 3) the observed long-term sequence similarity is not primarily due to universally conserved sites across all orthologs. Instead, as the reviewer correctly comments, most sites are on average constrained to a library of less than 4 amino acids. We also show that long-term sequence constraints can be partially explained by a site’s fitness effects of mutations and its distance to catalytic residues.

If it is, I think it is a meaningful finding that could be explained better to the readers. I guess the second claim is that 3-5 amino acid limit is universal to all enzymes and (conserved!) non-enzymes. I do doubt (as in the original review) the validity of such a strong claim, which could be a result of the authors' bias in selecting families for their analysis. At least for non-enzymes, they selected most conserved proteins (like ribosomal proteins), so of course such selection is biased to get proteins that saturate easily in evolution. The authors try to justify this biased selection suggesting that it is difficult to find orthologs. But that statement by itself totally discredits this study. Why? Because if you cannot find your orthologs, wouldn't it mean that they already diverged beyond (lower than) your claimed 25% identity as the universal limit, and the author's conclusions do not apply to such families? Maybe for the enzymes too, the 25% limit is simply a reflection of the search methods the authors used to find orthologs, that fail to find more distant ones.

We apologize for the possible confusion. But we discussed this particular concern in the manuscript, and are afraid that the related discussion escaped the attention of the reviewer. First, for almost all analyzed enzymatic protein families, sequence divergence saturates at a relatively high sequence identity (25%-40%); at these sequence identities, protein orthologs can be easily identified by computational sequence comparison methods. Second, when we relax the stringency of the molecular function conservation (e.g. from sharing all digits of an EC number (EC4) to sharing only the first three digits (EC3)), we immediately observe orthologous protein pairs with significantly lower levels of sequence identity (Figure 6). For non-enzymatic protein families, we need to investigate families that conserved specific molecular functions across billions of years of evolution. That is why we are limited in the number of protein families we can consider, not because we cannot find orthologs, but because there are not many such non-enzymatic protein families (beyond known examples such as ribosomes, flavoproteins or chaperones) that continuously conserved specific molecular functions across billions of years of evolution.

To improve the presentation and make this paper quite interesting (well, reviewers are not supposed to direct the study, but the authors seem passionate about their work and also seem rather inexperienced in both logical thinking and putting a paper together, so maybe this recommendation could be helpful), I would suggest to base it on two plots.1) Between-protein variability.The first plot is the histogram of the average estimated number of amino acids allowable per site (without invariant sites, or with) for enzyme families and other protein families. Well, the authors would have to try harder to find orthologs for proteins that manage to diverge below 25%, even my rotation students can do that. This histogram would be expected to have a maximum around 3 to 5, and for some variable proteins it could be around 10 or 15? Right? Then protein families with very low and very high numbers could be discussed with an attempt to give explanations about their uniqueness.2) Within-protein variability.The second plot is the histogram of estimated number of amino acids allowable per site within a protein family. The authors could either normalize to the average per protein, or select a bin with the maximum count from previous plot (let's say 4). These histograms can be averaged for all families with mean and SD showing for each bin. I would assume there will be a large count for invariant sites for enzymes (at 1), counts for 2 amino acids used, 3, 4, etc. Will this histogram have a single mode? Maybe around 4? Several modes? Will there be sites using more than 10 amino acids? How are these distributed in spatial structure and relative to the active site? Discussion of these details could be quite insightful and interesting.If these authors do not wish to make these plots, since this review will be published, maybe someone else will, and we will learn something interesting as a result.

We thank the reviewer for the suggestions, but would like to first make an editorial comment. We do not think that condescending personal comments, i.e. calling into question the authors’ abilities for logical thinking (see more about it below), are either appropriate or helpful. We do not take these comments personally, but anonymously throwing personal comments is not what science or publishing should be about. We hope that the reviewer agrees. We also believe that publication of peer-reviews by *eLife* and other prominent journals will raise the overall level of scientific discussion, and make reviewers feel responsible for their comments.

Back to science. The comments above are especially surprising because the suggested analyses were mostly already present in the manuscript. We again think they may have simply escaped the reviewer’s attention. We have now made several further changes to highlight these results:

1) The between-protein variability in long-term sequence identity (previously Figure 1—figure supplement 2A) is now presented as a main figure in the manuscript (Figure 2A). We have added an additional x-axis (at the top of the figure) to show the corresponding average number of accepted amino acids per site; we also now present the results of an equivalent analysis for non-enzymatic protein families (Figure 2B).

2) In regards to the comparison between enzymes that conserve their function and proteins that “manage to diverge below 25% identity”, we have now modified Figure 6A to highlight the higher average number of acceptable amino acids for ancient orthologs sharing only three digits of the EC annotation. We also show the sequence identity of proteins with different molecular functions but the same structural fold. Additionally, we compared those differences to the corresponding long-term structural similarities between orthologs (Figure 6B).

3) In terms of within protein variability, we now comment on the different shapes of the distributions shown in Figure 3—figure supplement 1. We decided not to average these distributions across enzymes, as this would hide an interesting underlying variability in the number of acceptable amino acids across protein families.

As we describe in the manuscript (see discussion of Figure 6), the aforementioned results demonstrate that the average number of acceptable amino acids per site increases from the case when a specific molecular function is conserved (~3 amino acids for EC4), to conservation of less specific functions (~4 amino acids for EC3), to simply preserving the same protein fold (~10 amino acids). Naturally, a higher long-term sequence identity limit usually corresponds to a smaller average number of acceptable amino acids per site.

Currently, there are so many plots in this paper and many of them are not particularly helpful to get the point across quickly. Also I still find the usage of non-standard terms (like divergence rate) confusing, not necessary, and not insightful about the mechanism. Yes, the terms are defined, but they are just masking the reality, which is simpler: constrained usage of amino acids in positions of enzyme molecules. Not all 20, not even 10, but 4! Why not state and illustrate this clearly? Don't you agree that the impact of the paper will increase because it will be easier to understand?

We appreciate the reviewer’s opinion. As we described above, we now further highlight the aforementioned result about the number of acceptable amino acids. Nevertheless, as we stated previously, this is not the only main result in the paper. There are many other interesting results, analyses, and experiments described in the manuscript. We understand that the reviewer particularly appreciates this result. We do too! But, based on the comments of other reviewers and editors, and discussions with our colleagues, many researchers are likely to be interested in other results as well. Such as the existence of the sequence and structural divergence limits, temporal profiles of long-terms divergence, various structural and functional analyses, correlations with experimentally measured fitness effects, etc. Therefore, we cannot reduce the paper to one particular result or figure. We currently have 6 main figures in the manuscript, which is arguably not many, and, we believe, they can be easily understood by most readers interested in the topic. We hope that all of our analyses are useful and provide different and complementary insights to different researchers.

In terms of the “divergence rate”, we do not understand this comment. The terms “divergence rate” or “protein divergence rate” are in fact quite standard and popular in molecular evolution. Searches for these terms in Google or PubMed return many hundreds of usages in the same or similar context and in multiple related papers. We now also define it in the text, to prevent any possibility for misinterpretation or confusion.

The Abstract is still very poor and misleading. For instance, the statement "The effective divergence limit (>25% sequence identity) is not primarily due to multiple substitutions at the same sites" is completely wrong. According to the authors' explanations, this "effective divergence limit" is exactly due to multiple substitutions at the same site! If you have only 4 amino acids acceptable at a site, due to multiple substitutions sequence identity will saturate are 25%. Like in DNA. Why not write a precise and clear Abstract about what this work presents? Due to so many logical flaws in the authors' thinking and presentation (as illustrated above), I would be scared to publish this paper without a careful read. One statement at a time. And trying to figure out why the statement is wrong. If clearly not wrong, then move on to the next one.

We apologize for the possible confusion, but we do not believe that our statements are either wrong or illogical. Let’s consider the example given by the reviewer. It is true that DNA sequence identity saturates at 25% under the assumption that all nucleotide substitutions are equally accepted at DNA sites. In this case the limit is indeed directly driven by random and neutral substitutions. But in the case of proteins, this scenario, i.e. acceptance of all amino acids with the same probability, would lead to saturation at 5% sequence identity, and not at 25-40%, which is what we observe for proteins with conserved molecular functions. That is why we said that “the limit is not *primarily* driven by multiple substitutions at the same sites”. Obviously, substitutions at the same sites occur all the time, but the small number of acceptable substitutions is ultimately *not the cause but the consequence* of the functional constraints and the divergence limit. In any case, to eliminate any possibility of confusion here, following the reviewer’s suggestion, we rewrote the Abstract accordingly.

And, finally, why not compare your results with this paper more thoroughly PMID: 27138088? Isn't it a bit similar? I guess I don't understand the meaning behind "to investigate the temporal patterns of the long-term divergence."Do the authors still claim that the sites are saturated at the usage of 3 to 5 amino acids because the time passed was not sufficient to gain more changes? Or the time was enough and protein of the same function simply cannot tolerate additional amino acids well and still keep the function? Which one is right? I got an impression it was the latter. And then what I said at the beginning of this review holds. If it is the former, it needs to be convincingly justified.

We apologize for possible confusion. By “the temporal patterns of the long-term divergence” we mean the changes in sequence and structural similarities between orthologs as a function of time. As we discuss in the paper (see trajectories in Figure 1 and Figure 5A), our results clearly demonstrate that most ancient orthologs are close to saturation, and are unlikely to diverge, at least on planetary timescales, beyond ~25% sequence identity.

In terms of the paper by Jack et al., as we state in the paper, Jack et al. explored the difference in short-term substitution rates as a function of sites’ distance to catalytic residues. In contrast, our study is focused on the global long-term sequence and structural divergence across billions of years of evolution, the observed divergence limit, its origins, and the related analyses, such as contribution of various sites. Importantly, as we already discussed in our previous reply, the differences in short-term evolutionary rates do not imply the existence of a divergence limit, do not inform about the timing at which the divergence of orthologs saturates, or explain the functional origins of this saturation.

If the authors disagree with my review, then I did not understand the paper, which is possible, and still suggests that the authors need to improve the presentation.

We believe that the reviewer understood the paper, and thank the reviewer again for the comments and suggestions.

Reviewer #3:

This version of the manuscript is a significant improvement over the previous version in terms of added details (e.g. experimental procedure of folA mutagenesis re now described in sufficient detail to be understandable and/or potentially reproducible).The authors attempted to address my (and other reviewers) main concern by presenting the analysis in new Figure 5 that shows that full orthologues (all EC numbers coincide) are more sequence-constrained than partial orthologues (3 EC numbers coincide). However this analysis fails to control for different structural divergence between full and partial orthologues.Therefore, the most problematic aspect of the analysis – that authors attribute observed sequence conservation in diverging clades to conservation of function between orthologues has not been fully addressed. A clear alternative explanation that such conservation is explained by maintenance of structure and stability regardless of function still stands.The authors used a truism by pointing out that proteins of the same FOLD (i.e. topologically similar arrangement of elements of secondary structure) can diverge to 10% or less sequence ID again citing an old work with data collected on very limited number of structures available at that time.However, their full functional orthologues are much more similar structurally than proteins sharing the same fold. The correct control which was suggested in my initial review has not been done satisfactorily. Specifically, the authors should compare their sets of proteins with sets that have similar degree of structural similarity (measured as distribution of TM-scores) but different function. There are such examples where structures are quite similar (TM-score-wise) but functions differ significantly. Good examples of that kind ate TIM-barrels which are almost exclusively enzymes with wide variety of specificity and Igfold proteins again with very conserved structures but broadly diverged functional annotations.If the authors find that conservation of structure in functionally divergent proteins imposes less sequence divergence constraints than same degree of structural conservation in orthologues – that will be a clear demonstration of additional constraints imposed by functional conservation which is the main message of this work. In the absence of such analysis the current data does not justify the conclusion.

Following the reviewer’s comments, we now present the suggested analysis (Figure 6C) testing the contribution of function to sequence constrains independent of structural similarity. The results clearly demonstrate that, even at the same level of structural divergence (RMSD), there is significantly higher sequence similarity for pairs of orthologs sharing the full EC annotation, i.e. a higher degree of functional similarity. We have updated the Discussion to reflect this new result.

Importantly, we would like to emphasize that the main message of our work is not Figure 6C, but the fact that the conservation of specific molecular function limits how far orthologs can diverge both in sequence and structure (i.e. in terms of the RMSD). As we comment in the Discussion, this limit is likely due to the conservation of structural optimality (but not necessarily protein stability, see below), i.e. of the specific structural arrangements and dynamics of protein residues required for efficient catalysis and function. To further emphasize this point, we now show side by side (Figure 6A, 6B) the long-term sequence and structural divergence of orthologs sharing 1.) their full EC numbers, 2.) orthologs sharing only the first three digits of their EC numbers, and 3.) enzymes classified in the same structural fold, but with different molecular functions. These results clearly demonstrate that orthologs maintaining a less stringent level of functional conservation diverge significantly further in both sequence and structure.

Minor point: The authors severely misquote Firnberg et al., 2016. They say: "Nevertheless, direct and comprehensive biochemical experiments demonstrated that the deleterious effects of protein mutations primarily result from changes in specific protein activity rather than decreases in protein stability and cellular abundance [Firnberg et al., 2016]". In fact, the authors of Firnberg et al., 2016, say directly the opposite: '… These DFEs provide insight into the inherent benefits of the genetic code's architecture, support for the hypothesis that mRNA stability dictates codon usage at the beginning of genes, an extensive framework for understanding protein mutational tolerance, and evidence that mutational effects on protein thermodynamic stability shape the DFE..…" (cited from the Abstract of Firnberg et al., 2016). The authors misrepresent the main result of Firnberg et al., 2016: According to Figure 5B of [Firnberg et al., 2016] the product of abundance and catalytic activity shapes fitness effects in TEM1, not abundance alone. This is exactly what is established in Bershtein et al., 2015 and Adkar et al., 2017, and shows that abundance (which is a function of stability) enters fitness landscape on equal footing with catalytic activity, i.e. there is as much selection for abundance (i.e. stability) as it is for kcat/KM and related measures of activity. This misinterpretation somewhat undermines the major premise of the present paper that there is separate selection for activity/function and stability/structure. In reality one cannot disentangle the two because fitness landscape depends on the function (i.e. the product) of the two factors.

Frankly, we are quite surprised by the Reviewer’s comment. The reviewer suggests that we misquote the Firnberg *et al.* study and cites several sentences from their abstract. We are puzzled by this comment because immediately after the cited text, in fact in the next sentence (!) of the abstract, Fimberg *et al.* directly and clearly state: “*Contrary to prevailing expectations, we find that deleterious effects of mutations primarily arise from a decrease in specific protein activity and not cellular protein levels*”, which is exactly what we are saying in our paper.

Of course we do not dispute the fact that Fitness ~ Enzyme_Rate= kcat*[E] (under ligand saturation), which is textbook biochemical knowledge. Also we do not dispute that stability effects *do contribute* to protein evolution, we specifically said that in the text and previous reply. However, it is simply not true that because Fitness(x)=kcat(x)*[E(x)] (where “x” a specific mutation), it is not possible to understand the average relative contribution of random mutations to changes in fitness due to functional (kcat) or stability/abundance ([E]) effects. In direct mathematical terms: if P(x)=A(x)*B(x) it is clearly possible to disentangle the relative contribution of A and B to the variance in P due to changes in x. In fact, the experiments in Figure 5 of Firnberg *et al.* were specifically designed to do that, and they directly support the primary contributions of effects related to protein activity. The different, and in fact sometimes opposite, contributions of stability and activity to protein evolution are also illustrated by several examples of enzymatic optimization where protein stability is traded for an increase in catalytic efficiency (see for example PMID: 26940154, PMID: 20036254).

In our view, the discussion above is not a minor point. These results support our conceptual model of the observed limit. We also think the Reviewer maybe is confused that mutations can affect protein function (i.e. kcat) only through purely sequence effect. In fact, sequence substitutions affect the average protein structure and dynamics (but again not necessarily protein stability!) and make the structure not optimal for catalysis and function. Furthermore, we would like to emphasize again that there is no specific selection in evolution for protein stability, i.e. evolution does not just select proteins to be stable *per se*. Obviously, proteins need to have a certain level of stability for functioning, but, based on the Fimberg *et al.* results, the selection for protein stability is not the only and likely not the main target of functional selection.